# VERITRAIL: CLOSED-DOMAIN HALLUCINATION DETECTION WITH TRACEABILITY

**Dasha Metropolitansky, Jonathan Larson**
Microsoft Research
{dasham,jolarso}@microsoft.com

## ABSTRACT

Even when instructed to adhere to source material, language models often generate unsubstantiated content – a phenomenon known as "closed-domain hallucination." This risk is amplified in processes with multiple generative steps (MGS), compared to processes with a single generative step (SGS). However, due to the greater complexity of MGS processes, we argue that detecting hallucinations in their final outputs is necessary but not sufficient: it is equally important to trace where hallucinated content was likely introduced and how faithful content may have been derived from the source material through intermediate outputs. To address this need, we present VeriTrail, the first closed-domain hallucination detection method designed to provide traceability for both MGS and SGS processes. We also introduce the first datasets to include all intermediate outputs as well as human annotations of final outputs' faithfulness for their respective MGS processes. We demonstrate that VeriTrail outperforms baseline methods on both datasets.

## 1 INTRODUCTION

Language models (LMs) are widely used to generate content based on source text. However, even when instructed to adhere to the source, LMs are known to produce unsupported content – a phenomenon known as "closed-domain hallucination" (OpenAI et al., 2024). Detecting hallucination is important in many settings (e.g., a doctor seeking guidance from medical literature, a lawyer summarizing precedent cases, a customer service agent answering questions based on policy documents). Closed-domain hallucination detection is also known as "faithfulness evaluation" – we use both terms (as well as "hallucination detection" for brevity) interchangeably throughout this paper.

Processes that use LMs to generate content based on source material can be divided into two categories: processes with a **single generative step (SGS)** and processes with **multiple generative steps (MGS)**. In SGS processes, the LM produces only a final output, without generating any intermediate outputs. One limitation of SGS processes is that they may be less reliable for long documents or large collections of documents: if the source material exceeds the model's context window, truncation or retrieval is required, increasing the risk of information loss.

Given the limitations of SGS processes, applications of LMs increasingly rely on MGS processes, where intermediate outputs generated by the LM are used as inputs to subsequent steps. In MGS processes, since the source material (or the task) can be split into smaller parts, truncation or retrieval may not be necessary, reducing the risk of information loss. However, MGS processes are more susceptible to hallucination, as each step presents an additional opportunity for errors to arise and propagate. Therefore, as the use of MGS processes accelerates, effective hallucination detection methods become increasingly important.

We argue, however, that detection alone is not enough. In many settings, we need to understand *how* the output may have been derived from the source (**provenance**) and *where* errors may have been introduced (**error localization**). Provenance helps users verify and trust the output, while error localization is critical for addressing hallucinations and understanding which parts of the process are most error-prone. We refer to provenance and error localization collectively as **traceability**. The transparency enabled by traceability is especially important for MGS processes due to their complexity.

Prior works frame faithfulness evaluation as assessing whether a given LM output is supported by the source material and do not distinguish between intermediate and final outputs (see § 5 for examples). For SGS processes, this simplification is reasonable: since there is only one generative step, evaluating its output against the source text is sufficient both to assess faithfulness and to provide traceability. In MGS processes, however, while we can still evaluate the final output against the source text to determine whether hallucination occurred, we cannot achieve traceability without utilizing the intermediate outputs.

A simplistic approach to traceability is to check the final output against each individual intermediate output. However, this approach can be prohibitively expensive when there are many intermediate outputs (e.g., one of the processes studied in this paper produced over 100,000 intermediate outputs). The simplistic approach also fails when the final output is based on a combination of intermediate outputs. For instance, suppose the final output states: *"Company X acquired two startups in 2020 as part of its expansion into healthcare."* One intermediate output mentions a single acquisition in 2020; a second references another acquisition in 2020; and a third describes Company X's overall acquisition strategy in 2020 as focused on expansion into healthcare. Taken individually, none of these intermediate outputs supports the claim – but together, they do. These limitations highlight the need for an approach to traceability that goes beyond evaluating outputs in isolation.

This paper makes the following contributions:

1. We propose a conceptual framework that provides a unified representation of generative processes for the purpose of faithfulness evaluation.

2. We introduce VeriTrail, the first closed-domain hallucination detection method to provide traceability for MGS and SGS processes. We also demonstrate that VeriTrail outperforms baseline methods in hallucination detection while remaining cost-effective.

3. We construct FABLES+ and DiverseSumm+, the first datasets to include all intermediate outputs as well as human annotations of final outputs' faithfulness for their respective MGS processes.[1]

## 2 CONCEPTUAL FRAMEWORK

We define a **non-generative step** as an operation that outputs only unmodified text spans from the input (e.g., noun phrase extraction using spaCy; Honnibal et al., 2020). In contrast, a **generative step** may modify the input text or introduce new information. Steps involving LMs are typically generative, although exceptions exist (e.g., constrained decoding; Geng et al., 2024).

A **generative process** is a sequence of steps that produces a final output from a set of source documents $D$ and includes at least one generative step. At each step, the input consists of text spans from $D$ and/or outputs from earlier steps.

Generative processes can be categorized as containing either a **single generative step (SGS)** or **multiple generative steps (MGS)**. An example of an SGS process is Retrieval-Augmented Generation (RAG) where a non-generative retrieval system selects a subset of the source material to provide as input to an LM (Lewis et al., 2020). Although RAG involves two steps, it is an SGS process because only one step is generative. We provide examples of MGS processes in Appendix A, including two processes used in our experiments: hierarchical summarization (Wu et al., 2021; Chang et al., 2023) and GraphRAG (Edge et al., 2025).

We model a generative process as a directed acyclic graph (DAG) $G = (V, E)$, where each node $v \in V$ represents a text span, either originating from $D$ or produced by a step. Each directed edge $(u, v) \in E$ indicates that node $u$ was included as an input to the step that produced node $v$.

For any node $v$, we define its **source nodes** as $\mathrm{src}(v) = \{u \in V \mid (u, v) \in E\}$ – the set of nodes used as input to produce $v$. **Root nodes** $V_0 \subseteq V$ have no incoming edges (i.e., $\forall\, v \in V_0$, $\mathrm{src}(v) = \emptyset$) and correspond to text spans from $D$. The **terminal node** $v^* \in V$ has no outgoing edges and represents the final output. We refer to any node that is neither a root node nor the terminal node as an **intermediate node**.

---

[1]The datasets will be released at `https://aka.ms/veritrail-datasets`.

We define a function $\text{stage}: V \to \mathbb{N}$ that assigns a **stage** to each node, such that for each edge $(u, v) \in E$, $\text{stage}(v) \geq \text{stage}(u)$. Conceptually, the stage reflects a node's position in the generative process. Root nodes are assigned the minimal stage, and the terminal node is assigned the maximal stage. For intermediate nodes, some processes (e.g., GraphRAG) have clearly defined stage assignments due to distinct step types, while others (e.g., hierarchical summarization) do not have a single "correct" assignment. Appendix B.1.1 and Appendix B.2.1 describe the stage assignment procedures used in our experiments.

**Faithfulness evaluation** aims to determine whether the terminal node $v^*$ is supported by the root nodes that have a path to $v^*$. Intuitively, these root nodes represent the subset of the source material that could have contributed to the final output. To enable fine-grained evaluation, we follow prior work (e.g., Min et al., 2023; Hu et al., 2025) in decomposing the final output into a set of factual claims $C = \{c_1, \ldots, c_n\}$, where each claim is a self-contained, verifiable statement. Each claim $c \in C$ is assigned one of the following **verdicts**:

1. **Fully Supported:** The source text strongly implies the entire claim. A careful reader would naturally infer the claim without relying on assumptions or external knowledge.

2. **Not Fully Supported:** At least one part of the claim is not strongly implied by the source text. This may occur because the source text contradicts the claim, strongly implies it is false, only weakly implies it, or does not address it at all.

3. **Inconclusive:** The source text is ambiguous or conflicting such that both "Fully Supported" and "Not Fully Supported" verdicts are possible, with neither clearly favored.

## 3 VERITRAIL

In this section, we describe VeriTrail, our new closed-domain hallucination detection method.

### 3.1 HALLUCINATION DETECTION PROCESS

VeriTrail has three main inputs:

1. A DAG representing a completed generative process (as described in §2), where each node is assigned a unique ID;

2. A hyperparameter $q$, which specifies the number of consecutive "Not Fully Supported" verdicts that will trigger termination of the hallucination detection process; and

3. A set of factual claims $C$, extracted from the terminal node $v^*$ of the DAG.

In our experiments, we obtain $C$ by applying Claimify (Metropolitansky & Larson, 2025), a claim extraction method, to $v^*$. VeriTrail evaluates each claim $c \in C$ independently from the other claims.

The subsections below describe the main steps in VeriTrail's hallucination detection process, which assigns a verdict to each claim. An example is shown in Figure 1, and the full procedure is provided in Algorithm 1 in Appendix C.1. All prompts are in Appendix C.2.

### 3.1.1 SUB-CLAIM DECOMPOSITION

A claim may contain multiple distinct parts, which we refer to as "sub-claims." For example, the claim *"Company X acquired two startups in 2020 as part of its expansion into healthcare"* can be split into: (1) Company X acquired two startups in 2020, and (2) these acquisitions were part of its expansion into healthcare. Identifying sub-claims is important because, as noted in §2, all parts of a claim must be supported in order to justify a "Fully Supported" verdict.

To identify sub-claims for the claim $c$, VeriTrail applies Claimify's Decomposition module, which attempts to rewrite the input text as a set of simpler, independently verifiable statements. If multiple sub-claims are returned, they are added to a queue for further decomposition. If only a single sub-claim is returned, it is treated as final and not processed further. To prevent infinite loops, VeriTrail skips previously processed sub-claims and enforces a maximum number of decomposition attempts.[2] Sub-claims are retained as context for the following steps but are not verified directly.

---

[2]We set this maximum to 20 in our experiments.

### 3.1.2 EVIDENCE SELECTION

Next, VeriTrail identifies $src(v^*)$ – the set of nodes used as input to produce the terminal node. Each source node is segmented into sentences using NLTK's sentence tokenizer (Bird & Loper, 2004), and each sentence is programmatically assigned a unique ID. An LM is then prompted to select all sentences that strongly imply the truth or falsehood of $c$ or any of its sub-claims (see the prompt in Appendix C.2.1 for details).

If the sentences do not fit within a single prompt, they are split across multiple prompts, which are processed in parallel to minimize latency. By default, each prompt includes as many sentences as can fit within the LM's context window, although this limit is configurable. See Appendix E.2 for an ablation study on the effect of the input size limit.

The LM returns the IDs of selected sentences and a summary of their combined content. If a returned ID does not match a programmatically assigned ID, it is discarded; otherwise, it is mapped to its corresponding sentence. This approach guarantees that the sentences included in the evidence trail are not hallucinated. Additionally, identifying specific sentences that support or refute the claim is arguably more informative than classifying entire nodes as relevant or not. It also simplifies human verification, since reviewing a selection of relevant sentences requires significantly less effort than reading full passages.

### 3.1.3 VERDICT GENERATION

If no sentences are selected during Evidence Selection, the claim $c$ is assigned a "Not Fully Supported" verdict. Otherwise, an LM is prompted to assign one of three verdicts ("Fully Supported," "Not Fully Supported," or "Inconclusive") based on the Evidence Selection results (see Appendix C.2.2).

Sentences selected during Evidence Selection are not included directly in the Verdict Generation prompt, as they may be ambiguous when used out of context. For example, if the claim is *"John Smith was the CEO of Company X,"* and a selected sentence is *"He served as its CEO from 2006-2010,"* it is unclear whether *"He"* refers to John Smith and *"its"* to Company X. To avoid such ambiguities, VeriTrail determines the input for the Verdict Generation prompt based on the nodes from which the sentences were selected: if the node is a root node, its full content is included; otherwise, the summary generated during Evidence Selection is used.

As in Evidence Selection, an input size limit can be specified for the Verdict Generation prompt. However, unlike in Evidence Selection, Verdict Generation requires all inputs (i.e., the evidence) to fit within a single prompt. Therefore, if the input size limit is exceeded, VeriTrail reruns Evidence Selection – this time on the previously selected evidence – until either (a) the evidence fits within the limit, or (b) a maximum number of reruns is reached (where this maximum is configurable). If condition (b) occurs, then the largest subset of evidence that fits within the limit is used.

### 3.1.4 CANDIDATE NODE SELECTION AND TERMINATION

Recall from §3.1.2 that VeriTrail selects the source nodes of the terminal node, $src(v^*)$, as input for the initial round of Evidence Selection. Once Evidence Selection and Verdict Generation based on $src(v^*)$ are complete, VeriTrail selects a new set of nodes to use as input for the next round of Evidence Selection. For ease of reference, we call this set the "candidate nodes." The candidate nodes are selected as follows:

- If the latest verdict was "Fully Supported" or "Inconclusive": include the source nodes of all nodes from which sentences were selected during the latest round of Evidence Selection.

- If the latest verdict was "Not Fully Supported": include the source nodes of all nodes verified in the latest iteration – not just those that yielded evidence.[3]

---

[3]In the "Not Fully Supported" case, the primary motivation for verifying the source nodes of all previously verified nodes, regardless of whether they yielded evidence, is to reduce the risk of false positives (i.e., incorrect "Not Fully Supported" verdicts). False positives can be caused by Evidence Selection failing to select evidence from nodes that support the claim. If only the source nodes of evidence-yielding nodes are considered for further verification, nodes overlooked by Evidence Selection will be excluded from future iterations, perpetuating the error. By including the source nodes of all previously verified nodes, VeriTrail increases the likelihood of recovering missed supporting evidence for the claim.

- Include any root nodes from which evidence was selected in previous iterations. These nodes are not reprocessed during Evidence Selection but are used in Verdict Generation.[4]
- To avoid redundant processing, exclude any candidate nodes that were verified in earlier iterations, except for the root nodes described above.

The hallucination detection process terminates if any of the following conditions is met:

1. The candidate nodes consist only of previously verified root nodes from which evidence was selected;
2. There are no candidate nodes, meaning that the root nodes were never reached or that none of the root nodes yielded evidence; or
3. The number of consecutive "Not Fully Supported" verdicts has reached $q$.

Under condition 1, the latest verdict is deemed final. Under conditions 2 and 3, the final verdict is set to "Not Fully Supported." If none of these conditions is met, the steps described in §3.1.2, §3.1.3, and §3.1.4 are repeated using the newly identified candidate nodes as input for the next round of Evidence Selection.

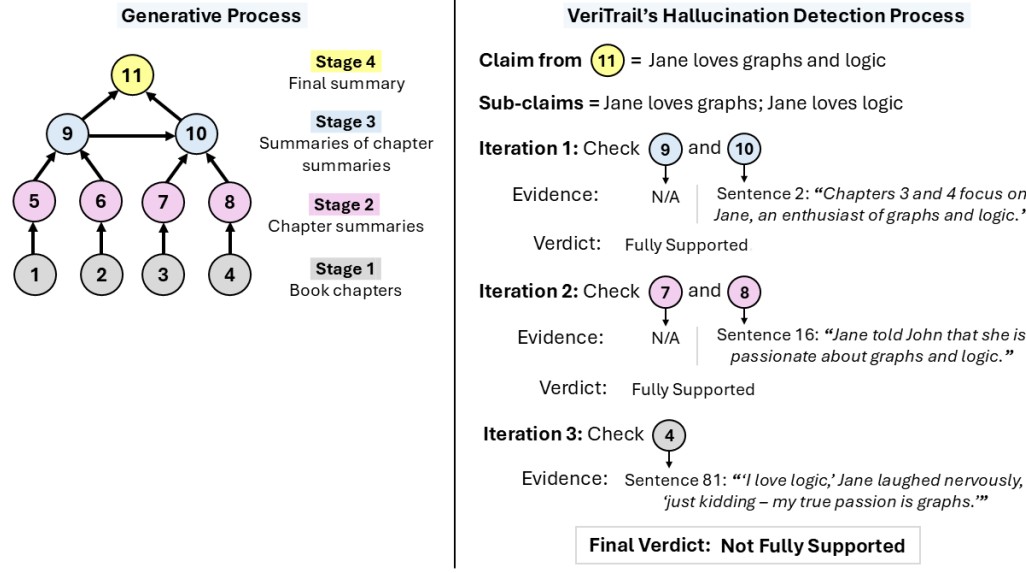

**Figure 1:** Left: Hierarchical summarization as a DAG. Circles represent nodes; arrows represent edges. Right: VeriTrail's hallucination detection process. The evidence trail includes sentence 2 from node 10, sentence 16 from node 8, and sentence 81 from node 4. Evidence summaries are not shown.

## 3.2 TRACEABILITY

For each claim extracted from the terminal node $v^*$, VeriTrail returns: (1) a final verdict, along with the LM's reasoning; (2) all interim verdicts; and (3) an evidence trail composed of (a) selected sentences with their corresponding node IDs and (b) generated summaries from all iterations of Evidence Selection.

Collectively, these outputs provide traceability as follows:

---

[4]For example, recall the claim *"Company X acquired two startups in 2020 as part of its expansion into healthcare."* Suppose that in the latest iteration, the claim was deemed "Fully Supported" based on (1) an intermediate node stating that Company X acquired two startups in 2020, and (2) a root node stating that Company X's acquisition strategy focused on healthcare expansion. In the next iteration, we need to verify the source nodes of the intermediate node. Assume the source nodes contain the same information as the intermediate node. If we exclude the already-verified root node from Verdict Generation, the information about healthcare expansion would be lost, and the claim would be incorrectly labeled "Not Fully Supported."

- **Provenance.** For claims whose final verdict is "Fully Supported" or "Inconclusive," the evidence trail documents a path through the intermediate nodes to the root nodes.
- **Error Localization.** For claims deemed "Not Fully Supported," the interim and final verdicts are used to identify **error stage(s)** – the stage(s) where the unsupported content was likely introduced.

To find the error stages, VeriTrail identifies iteration $n$, the last iteration where the claim received an interim "Fully Supported" verdict prior to the final "Not Fully Supported" verdict. Let $V_e(n)$ denote the set of nodes from which at least one sentence was selected during the Evidence Selection step in iteration $n$. The error stages are then defined as the stages of the nodes from which sentences were selected during Evidence Selection in iteration $n$, excluding root nodes[5]: $\{\text{stage}(v) \mid v \in V_e(n), v \notin V_0\}$. For example, in Figure 1, the last iteration that received a "Fully Supported" verdict before the final "Not Fully Supported" verdict was Iteration 2. In Iteration 2, evidence was selected only from node 8, which belongs to stage 2. Therefore, the error stage is stage 2.[6]

There are three scenarios where a claim never receives a "Fully Supported" verdict. First, the first $q$ iterations returned "Not Fully Supported," causing verification to terminate. In this case, the error stage is $\text{stage}(v^*)$ (i.e., the unsupported content was introduced in the final output). Second, the interim verdicts were a mix of "Inconclusive" and "Not Fully Supported" verdicts. Third, all interim verdicts were "Inconclusive." In the second and third scenarios, an error stage cannot be identified.

## 4 EXPERIMENTS

### 4.1 DATA

Since detecting hallucinations in MGS processes is increasingly important yet underexplored, we focused on evaluating VeriTrail's performance across diverse MGS processes. With regard to the source material, we targeted long documents and large document collections (i.e., >100K tokens), where hallucination detection is especially challenging and MGS processes tend to be most valuable.

While many existing datasets provide human annotations of faithfulness for LM-generated outputs, few are based on MGS processes, and we did not find any that include all intermediate outputs. To address this gap, we constructed two new datasets by augmenting prior work:

1. **FABLES+** is based on FABLES (Kim et al., 2024), a dataset of book summaries generated via hierarchical summarization. Since the original dataset did not preserve all intermediate outputs required to construct the DAG input for VeriTrail, we re-generated summaries for 22 books with an average length of 118K tokens. We extracted 734 claims from the final summaries using Claimify. 48% of the extracted claims restated information that had already been deemed faithful in the original dataset. As a result, we labeled these claims as faithful without further annotation. We manually annotated the remaining claims.

2. **DiverseSumm+** is based on DiverseSumm (Huang et al., 2024), a dataset of news stories (e.g., the Russia-Ukraine conflict), each linked to 9-10 articles and a set of questions answered by multiple articles. We retained 148 stories, linked to 1,479 articles with a cumulative length of 1.19M tokens. From these stories, we sampled 20 questions and generated answers using GraphRAG over the full article set. We then extracted 560 claims from the answers using Claimify. To label the claims' faithfulness, we recruited four annotators through Upwork. Annotators were given access to the 9-10 articles associated with each claim, while one of the authors served as a fifth annotator with access to the full article set. For 87% of claims, the associated 9-10 articles were sufficient to determine faithfulness; 13% required consulting additional articles.

Full details for both datasets are provided in Appendix B.

---

[5]As defined in §2, the root nodes are the ground truth, so they cannot be a source of errors.

[6]VeriTrail may return multiple error stages. For example, in GraphRAG, the source nodes for certain stage 4 nodes may include both stage 2 and stage 3 nodes (see Appendix B.2.1 for stage definitions). If $V_e(n)$ includes nodes from both stages, VeriTrail lists both as potential error stages since it is not possible to attribute the hallucination to just one.

## 4.2 BASELINE METHODS

We compared VeriTrail against three types of methods commonly used to evaluate faithfulness and capable of processing large source texts:

1. **Natural Language Inference (NLI).** NLI methods assess whether a claim is entailed by a source document. We selected three methods that represent different strategies for handling large source documents:

   (a) **INFUSE** (Zhang et al., 2024a) splits the document into sentences and ranks them based on bi-directional entailment (i.e., the probability that the sentence entails the claim or vice versa). It constructs an evidence set by adding top-ranked sentences, stopping when the predicted probability that the claim is neutral with respect to the evidence set begins to increase. The entailment probability is computed over the final evidence set.

   (b) **AlignScore** (Zha et al., 2023) splits the document into small chunks (~350 tokens) and computes the probability that each chunk entails the claim. We aggregated these probabilities using the mean of the top-$k$ values, testing $k \in [1, 15]$.

   (c) **Llama-3.1-Bespoke-MiniCheck-7B** (Bespoke Labs, 2024) operates similarly to AlignScore, but it uses much larger chunks (~32K tokens). For each claim, we computed the mean of the top-$k$ entailment probabilities, testing $k \in [1, 5]$ for FABLES+ and $k \in [1, 15]$ for DiverseSumm+.

   To convert entailment probabilities into binary labels, we tested thresholds $\tau \in \{0.5, 0.6, 0.7, 0.8, 0.9\}$: a claim was labeled "Fully Supported" if the entailment probability was at least $\tau$, and "Not Fully Supported" otherwise. See Appendix F for details.

   Finally, we note that Zha et al. (2023) and Bespoke Labs (2024) reported that AlignScore and Llama-3.1-Bespoke-MiniCheck-7B outperformed much larger models (e.g., GPT-4 and Claude-3.5-Sonnet, respectively) on standard factual consistency benchmarks (e.g., SummEval, QAGS, and LLM-Aggrefact), making them particularly strong baselines.

2. **Retrieval-Augmented Generation (RAG).** We reused the document chunks created during dataset construction (see Appendix B.1.1 and Appendix B.2.1). Claims and chunks were embedded using OpenAI's `text-embedding-3-large` model. For each claim, we retrieved the top-$k$ most similar chunks, which were passed to an LM for verdict generation. For retrieval, we used Faiss' k-nearest neighbors search (Douze et al., 2025) with L2 distance, testing $k \in \{1, 3, 5, 10, 15, 20, 25, 30\}$.

3. **Direct Verification Using Long-Context LMs.** We provided the source document(s) – a full book for FABLES+ and all 1,479 articles for DiverseSumm+ – directly to an LM for verdict generation. We tested two models with long context windows: **Gemini 1.5 Pro** (2M tokens) and **GPT-4.1 Mini** (~1M tokens). Since the DiverseSumm+ articles exceeded GPT-4.1 Mini's context window, each time we evaluated a claim, we randomly shuffled the articles and selected the largest subset that fit (typically ~80%).

## 4.3 RESULTS

We evaluated VeriTrail and baseline methods in two settings: **hard prediction**, where methods output a single label per claim ("Fully Supported" or "Not Fully Supported")[7], and **soft prediction**, where methods produce a continuous score representing the probability that a claim is "Fully Supported." Soft prediction results are reported in Appendix E.3.

We report results for AlignScore, INFUSE, Llama-3.1-Bespoke-MiniCheck-7B, and RAG using the best-performing hyperparameter configuration for each dataset; results for all configurations are provided in Appendix E.4. For RAG and Direct Verification, we used VeriTrail's Verdict Generation prompt to ensure that any performance differences are not attributable to prompting. We also tested an alternative prompt from the FABLES paper, including for VeriTrail. As shown in Appendix E.5, the alternative prompt yielded worse results for all methods except GPT-4.1 Mini on FABLES+.

---

[7]Claims assigned an "Inconclusive" verdict by any method (4% for FABLES+ and 8% for DiverseSumm+) were excluded from the hard prediction evaluation. We also tested treating "Inconclusive" verdicts as either always correct or always incorrect and observed only marginal differences in the reported metrics.

Table 1 shows hard prediction results for FABLES+ and DiverseSumm+, respectively. Due to class imbalance (see Appendix B), we used macro $F_1$ and balanced accuracy as our primary metrics. For VeriTrail, we report results for $q = 1$ and $q = 3$; results for additional $q$ values are analyzed in Appendix E.6. All VeriTrail and RAG results in Table 1 were produced using OpenAI's `gpt-4o-2024-08-06` model; results for other models are included in Appendix E.7.

On our primary metrics, VeriTrail outperformed all baseline methods for both datasets and all models tested, except for `mistral-large-2411`, where VeriTrail had the highest balanced accuracy but not the highest macro $F_1$. VeriTrail also outperformed all baselines in the soft prediction setting (Appendix E.3).

**Table 1:** Hard prediction results (%) for the FABLES+ (F) and DiverseSumm+ (D) datasets. We report macro $F_1$, balanced accuracy (Bal. Acc.), and class-specific precision and recall for fully supported (FS) and not fully supported (NFS) claims. For RAG, AlignScore, INFUSE, and Llama-3.1-Bespoke-MiniCheck-7B (denoted "Bespoke-MiniCheck-7B"), we report the best-performing configuration by macro $F_1$: RAG uses $k = 15$ for both datasets; AlignScore uses $k = 1$, $\tau = 0.6$ for FABLES+ and $k = 1$, $\tau = 0.9$ for DiverseSumm+; INFUSE uses $\tau = 0.5$ for both datasets; Llama-3.1-Bespoke-MiniCheck-7B uses $k = 1$, $\tau = 0.5$ for FABLES+ and $k = 2$, $\tau = 0.5$ for DiverseSumm+. Bolded values represent the highest score in each column.

| Method | Macro $F_1$ | | Bal. Acc. | | Precision$_{FS}$ | | Recall$_{FS}$ | | Precision$_{NFS}$ | | Recall$_{NFS}$ | |
|---|---|---|---|---|---|---|---|---|---|---|---|---|
| | F | D | F | D | F | D | F | D | F | D | F | D |
| VeriTrail ($q = 1$) | 74.0 | 76.6 | **84.6** | **83.0** | **97.5** | 95.8 | 82.8 | 76.2 | 44.1 | 55.1 | **86.5** | 89.8 |
| VeriTrail ($q = 3$) | **84.5** | **79.5** | 83.6 | 76.3 | 95.4 | 87.1 | 96.4 | **96.7** | **75.6** | **84.5** | 70.8 | 55.9 |
| RAG | 69.6 | 75.1 | 76.5 | 74.0 | 94.6 | 86.7 | 83.3 | 90.5 | 39.6 | 66.4 | 69.8 | 57.5 |
| Bespoke-MiniCheck-7B | 62.2 | 72.1 | 69.0 | 69.4 | 92.6 | 83.8 | 77.7 | 95.4 | 29.9 | 75.3 | 60.4 | 43.3 |
| Gemini 1.5 Pro | 61.1 | 49.8 | 60.8 | 57.6 | 89.3 | 82.2 | 90.3 | 45.1 | 33.7 | 29.4 | 31.2 | 70.1 |
| GPT-4.1 Mini | 60.7 | 62.9 | 58.2 | 61.5 | 88.4 | 80.3 | **98.7** | 93.8 | 68.0 | 60.7 | 17.7 | 29.1 |
| AlignScore | 59.6 | 60.4 | 67.5 | 62.7 | 92.4 | 82.8 | 73.6 | 70.3 | 26.8 | 37.6 | 61.5 | 55.1 |
| INFUSE | 40.5 | 20.0 | 59.5 | 50.1 | 92.9 | **100.0** | 36.8 | 0.30 | 17.0 | 24.6 | 82.3 | **100.0** |

## 4.4 ANALYSIS

To supplement our main findings, we include the following analyses in the Appendix:

- Appendix D: We analyze VeriTrail's computational cost. We show that, despite a significantly larger verification burden, VeriTrail outperformed the baseline methods while remaining cost-effective.
- Appendix E.1: We present the results of an ablation study to identify which aspects of VeriTrail's design contributed most to its performance gains.
- Appendix G: We analyze error cases to assess VeriTrail's limitations.
- Appendix H: We examine the distribution of error stages identified by VeriTrail to understand where hallucinations tend to arise in the processes we studied.

## 5 RELATED WORK

**Open-Domain vs. Closed-Domain Hallucination.** Hallucination can arise in either a closed-domain or an open-domain setting. In a closed-domain setting, a model is given source material and is expected to remain faithful to it. In an open-domain setting, no source material is provided, so the model relies on its parametric knowledge when generating output. VeriTrail is designed specifically for the closed-domain setting.

**Reference-Free vs. Reference-Based Detection.** Hallucination detection methods can be categorized as reference-free or reference-based. Reference-free methods estimate the model's confidence in its output without consulting any reference text. Although traditionally used in open-domain settings, recent works have applied reference-free methods to closed-domain hallucination as well (e.g., Chuang et al., 2024; Sun et al., 2025; Ridder & Schilling, 2025). Because reference-free methods

rely on model internals (e.g., attention maps), they may not be usable for claims extracted from already-generated outputs, and they are often incompatible with closed-source models. In contrast, reference-based methods compare the model's output directly to a reference text, so they are model-agnostic and can be applied to already-generated outputs. In the open-domain setting, reference-based methods must retrieve reliable external evidence to serve as the reference text (e.g., Chen et al., 2024; Wei et al., 2024). In the closed-domain setting, the reference text is simply the source material that the model was given. A wide range of closed-domain, reference-based methods have been proposed, including fine-tuning models for question generation and answering (Durmus et al., 2020; Wang et al., 2020; Scialom et al., 2021) and using LMs as verifiers (Liu et al., 2023; Es et al., 2024; Saad-Falcon et al., 2024). VeriTrail is also a reference-based method.

**Evaluation of Intermediate Outputs.** To the best of our knowledge, no existing closed-domain hallucination detection method – reference-free or reference-based – accounts for the full structure of the generative process. Instead, prior methods evaluate LM outputs in isolation, which is insufficient to achieve traceability for MGS processes. A related area of work that involves the analysis of intermediate outputs is the evaluation of LM-generated reasoning chains (i.e., sequences of steps leading to a final answer). However, existing methods (e.g., Hao et al., 2024; Paul et al., 2024) focus on relatively simple chains (e.g., those that fit within a single prompt) and may not generalize to the longer, more complex processes addressed in this paper.

**Generative Processes as DAGs.** Several prior works have modeled generative processes as DAGs. In LangGraph (LangChain, 2025), nodes represent generative steps and edges denote execution order: an edge from $u$ to $v$ means $v$ was executed after $u$, but not necessarily that $u$'s output was used by $v$. Heterogeneous Swarms (Feng et al., 2025) adopts a similar structure, but requires that $u$'s output be used as input to $v$. In MacNet (Qian et al., 2025), both nodes and edges represent generative steps: an edge from $u$ to $v$ means $u$ produced an output, another step provided feedback, and $v$ refined the output based on the feedback. In VeriTrail's representation of generative processes, nodes represent text spans, not steps, and edges capture input-output relationships. For example, if an LM summarizes five document chunks in a single step, VeriTrail represents the process as six nodes – one for the summary, and one for each input chunk – rather than a single node.

# 6 CONCLUSION

In this paper, we address two increasingly important challenges: effective hallucination detection and traceability for processes with multiple generative steps. We introduce VeriTrail, the first closed-domain hallucination detection method that not only evaluates the output's faithfulness to the source material, but also provides provenance and error localization. For outputs deemed faithful, VeriTrail constructs an evidence trail that traces the path to the source material through intermediate outputs. For unfaithful outputs, it identifies the stages where hallucination likely occurred. As a result, VeriTrail provides transparency and actionable insights into generative processes.

To evaluate VeriTrail's performance and support future work on traceability, we created FABLES+ and DiverseSumm+, the first datasets to include all intermediate outputs of processes with multiple generative steps as well as human-annotated faithfulness verdicts. Across both datasets, which span diverse generative processes and source document types, VeriTrail outperformed strong baselines in hallucination detection while remaining cost-effective.

# 7 REPRODUCIBILITY

The paper includes all necessary information to re-implement VeriTrail, including detailed descriptions of the method and hyperparameters (§3), a complete algorithm specification (Appendix C.1), and all prompts (Appendix C.2). We also specify all settings for the baseline methods evaluated in our experiments (§4.2, Appendix F) to support reproducibility. Finally, we plan to release the FABLES+ and DiverseSumm+ datasets at `https://aka.ms/veritrail-datasets`.

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

## A    EXAMPLES OF PROCESSES WITH MULTIPLE GENERATIVE STEPS

We used the following processes with multiple generative steps (MGS) in our experiments:

1. **Hierarchical summarization** (Wu et al., 2021; Chang et al., 2023). Source documents are split into chunks. A language model (LM) summarizes each chunk individually, then the resulting summaries are repeatedly grouped and summarized until a final output is produced.

2. **GraphRAG** (Edge et al., 2025). Source documents are split into chunks. For each chunk, an LM extracts entities and relationships, along with short descriptions. If an entity or relationship was extracted from multiple chunks, an LM summarizes the descriptions. A knowledge graph is constructed from the final set of entities and relationships, then a community detection algorithm, such as Leiden clustering (Traag et al., 2019), groups entities into communities. For each community, an LM produces a "community report" that summarizes the entities and relationships. To answer a user's question, an LM generates "map-level answers" based on groups of community reports, then synthesizes them into a final answer.

Additional examples of MGS processes include:

- **Incremental Summarization.** Source documents are split into chunks, which are processed sequentially. An LM summarizes the first chunk; for each subsequent chunk, it updates the latest summary based on the current chunk (Chang et al., 2023; Hwang et al., 2024; Jayalath et al., 2025).

- **Indexing-Based Methods.** Many MGS processes include an indexing phase that is distinct from the querying phase. During indexing, the source material is converted into a structured representation, such as a graph or tree. Indexing is typically performed only once (unless the source material changes), while the querying phase occurs for each new query issued by the user. GraphRAG is one such method that uses a knowledge graph as its index structure. Additional examples include RAPTOR (tree-based; Sarthi et al., 2024) and HippoRAG (graph-based; Gutiérrez et al., 2024).

- **Multi-Agent Systems.** LM-based "agents," each responsible for a specific document or sub-task, process the source material in parallel or sequentially. A final agent synthesizes their outputs into a complete response. Examples include Chain of Agents (Zhang et al., 2024b), LONGAGENT (Zhao et al., 2024), and Mixture of Document Speakers (Balepur et al., 2025).

## B    DATASET DETAILS

This section provides additional details about FABLES+ and DiverseSumm+, the datasets we constructed by augmenting the FABLES (Kim et al., 2024) and DiverseSumm (Huang et al., 2024) datasets introduced in §4.1.

We performed claim extraction for both datasets using Claimify (Metropolitansky & Larson, 2025) with `gpt-4o-2024-08-06` and default hyperparameter settings. We manually removed redundant claims. The average number of claims per book in FABLES+ was 33, and the average per question in DiverseSumm+ was 28.

### B.1    FABLES+

#### B.1.1    OVERVIEW

The original FABLES dataset contains summaries of 26 books published from 2023 to 2024. The summaries were decomposed into claims, and each claim was labeled faithful or unfaithful by crowdworkers who had read the corresponding book. For FABLES+, we used 22 books for which Kim et al. provided access to the raw texts. The retained books' token counts ranged from 49,156 to 242,683, with an average of 118,092. Token counts were calculated using OpenAI's tiktoken library with the `cl100k_base` encoding.

To re-generate the summaries, we used the text chunks and hierarchical summarization implementation from the original paper. Our only modification was storing all intermediate outputs, which we later used to construct the DAG inputs for VeriTrail. We also obtained the original summaries of the text chunks from Kim et al. and re-generated only the higher-level summaries. We generated half of the summaries using OpenAI's `gpt-3.5-turbo` model and the other half using `gpt-4-0613`. Both models were also used in the original study, and we applied the same summarization parameters: $chunk\_size = 2,048$ and $max\_context = 8,192$.

When constructing the DAG for each book, we assigned stages to nodes as follows:

- Stage 1 = root nodes (i.e., book chunks);

- Stage 2 = nodes with at least one source node in stage 1 (i.e., summaries of book chunks);

- Stage 3 = nodes with at least one source node in stage 2, etc.

All DAGs had four stages: the root nodes, two intermediate stages, and the terminal node. The number of nodes in stages 1 and 2 ranged from 26 to 141, with an average of 67. The number of nodes in stage 3 ranged from 2 to 6, with an average of 2.7. The total number of nodes across all stages ranged from 29 to 148, with an average of 70.7.

To reuse as many labels as possible from the original dataset, we prompted an LM to check whether each extracted claim was entailed by a claim labeled as faithful in the original dataset or by any evidence or comments from the annotators, which we assumed to be faithful. We used `gpt-4o-2024-08-06` with the prompt in Appendix B.1.2. All matches were manually reviewed, and only clear cases were retained. For example, our extracted claim "*Altha is accused of the murder of John Milburn*" was deemed entailed by the following claim from the original dataset: "*The villagers and prosecutor believe Altha Weyward used her powers to make John Milburn's cows trample him to death, despite a lack of evidence.*" Since the claim from the original dataset was labeled as faithful in the original study, we classified our extracted claim as "Fully Supported."

Ultimately, 14% of claims in our dataset were labeled "Not Fully Supported," and the remaining 86% were labeled "Fully Supported." For summaries generated by GPT-3.5 Turbo, 12% of claims were labeled "Not Fully Supported" (88% "Fully Supported"); for GPT-4, 14% of claims were labeled "Not Fully Supported" (86% "Fully Supported").

### B.1.2 CLAIM MATCHING PROMPT

```
Claim Matching System Prompt

You are an expert in Natural Language Inference.
You will be given a premise and a hypothesis. Your task is to answer
the following: Given the premise, is the ENTIRE hypothesis
NECESSARILY TRUE? In other words, would it be correct to say that if
the premise is true, then the ENTIRE hypothesis MUST be true? If at
least one component of the hypothesis is NOT necessarily true based
on the premise, then the hypothesis is NOT necessarily true.

Note the following rules:
- You will NOT make any assumptions or speculations.
- You will NOT use any external information.
- You will NOT use any weak implications as the basis for the
hypothesis being necessarily true. Only strong implications are
allowed (note that this is a weaker standard than requiring explicit
statements).
- If the hypothesis consists of multiple components, ALL components
must be necessarily true given the premise in order for the
hypothesis to be necessarily true. For example, if the hypothesis is
"John works at Mary's favorite restaurant" and the premise is "John
works at a restaurant," then there is no evidence that the restaurant
 he works at is Mary's favorite restaurant.
```

---

**Claim Matching System Prompt (Continued)**

```
- You may also be given evidence and/or reasoning that helps explain
the premise. It is EXTREMELY important that you take it into account
when answering the question.

First, print the full premise and hypothesis. Then, identify all
components of the hypothesis.
Then, walk through your reasoning step-by-step. Remember: ALL
components of the hypothesis must be necessarily true for the
hypothesis to be necessarily true and weak implications, assumptions,
 and speculations are NOT allowed.
Lastly, print "Given the premise:", followed by one of the following:
 "The entire hypothesis is NECESSARILY TRUE" or "The hypothesis is
NOT NECESSARILY TRUE".
```

---

**Claim Matching User Prompt - FABLES+**

```
Premise:
Here is some information about the novel "{book}":
{premise}{evidence}{reasoning}

Hypothesis:
{hypothesis}
```

---

**Claim Matching User Prompt - DiverseSumm+**

```
Premise:
Here is an answer to the question "{question}":
{premise}{evidence}{reasoning}

Hypothesis:
{hypothesis}
```

## B.2 DIVERSESUMM+

### B.2.1 QUESTION SELECTION AND ANSWER GENERATION

The original DiverseSumm dataset contains 245 news stories. For each story, Huang et al. formulated questions answered by multiple articles with varied perspectives. As a result, DiverseSumm is a more realistic and challenging benchmark for hallucination detection than other datasets where questions have only a single answer from a single source. Moreover, since each question was linked to a single story, and the stories covered distinct topics, we expected the search space for faithfulness evaluation to be reasonably constrained – an important consideration for annotation reliability. To reduce article overlap across stories, we excluded stories with at least one article that paired with multiple stories.

We then filtered out questions that (a) asked for opinions (e.g., "*How might the SEC's issuance of the Wells Notice impact Coinbase's business processes?*") or (b) could not be understood without additional context (e.g., "*What assistance has been provided to the affected communities?*"). We sampled 20 of the remaining questions, each from a different story. We aimed to select a diverse set of questions covering a range of reasoning types. For example:

- Focused, factual: "*What is Genesis' strategy with the Electrified GV70?*"

- Broad, analytical: "*What are the long-term implications of the interest rates hike for the economy?*"

- Multi-hop: "*How will the IRS's approach to NFTs compare to its approach to other cryptocurrencies?*"

- Compound: "*What are the concerns surrounding user data and TikTok, and what legal measures have been put in place to address them?*"

To generate answers to selected questions using GraphRAG, we split the articles into 600-token chunks with 100-token overlap, resulting in 3,199 chunks. Token counts were calculated using OpenAI's tiktoken library with the `o200k_base` encoding. For the indexing phase of GraphRAG, we used OpenAI's `gpt-4o-2024-05-13` and `text-embedding-ada-002` models. For the querying phase, we used GraphRAG's `global_search` method with `gpt-4o-2024-08-06`.

To construct DAG for each question, we assigned stages as follows:

- Stage 1 = article chunks (i.e., the root nodes);
- Stage 2 = entities and relationships;
- Stage 3 = summarized entities and relationships;
- Stage 4 = community reports;
- Stage 5 = map-level answers; and
- Stage 6 = final answer.

The same indexing phase outputs (stages 1-4) were used for all questions. Only nodes in stages 5 and 6, produced during the querying phase, varied by question. The node counts for stages 1-4 were as follows: stage 1 = 3,199; stage 2 = 95,465 (entities = 43,125 and relationships = 52,340); stage 3 = 11,974 (entities = 5,584 and relationships = 11,974); stage 4 = 3,650. For stage 5, the number of nodes ranged from 27 to 170, with an average of 79. The total number of nodes across questions ranged from 114,316 to 114,459, with an average of 114,368.

### B.2.2 ANNOTATION PROCEDURE

To label the extracted claims, we recruited annotators through Upwork. As a screening test, candidates were asked to label 42 claims associated with one of the questions. We reviewed their labels and compared them to our own. Six candidates completed the test, and four whose results met our quality standards were selected to continue. All selected annotators were fluent English speakers, based in the United States, and had a 100% success rate for prior jobs. Three had completed a bachelor's degree. They were compensated at a rate of $15-20 per hour.

To minimize annotator fatigue, we divided the remaining samples into two batches, which were annotated sequentially over five days. Each annotator labeled all claims in both batches. The first batch included 6 questions, and the second included 13. We provided detailed feedback to the annotators after both the screening test and the first batch. To avoid overloading the annotators, they were shown only the 9-10 articles associated with each claim. Independently, one of the authors annotated all claims with access to the full set of articles. As noted in §4.1, only 13% of claims required the full set of articles, confirming our hypothesis that the constrained set would be sufficient to verify most claims.

Appendix B.2.3 contains the annotation instructions provided to the annotators. For the annotation interface, we created a separate Excel file for each question and annotator. The files contained one row per claim and columns for assigning a label, indicating uncertainty, quoting supporting or refuting evidence, and providing comments.

The label options used in the annotation study were more granular than those used by VeriTrail. Our goal was to encourage annotators to be as precise as possible about the relationship between each claim and the articles. Specifically:

- The labels "At Least One Part is Refuted" and "Insufficient Evidence (None of the Above)" were both mapped to VeriTrail's "Not Fully Supported" label, but the former required annotators to provide evidence that refuted the claim, while the latter indicated a lack of both supporting and refuting evidence.
- The "All Parts are Supported" label was mapped to VeriTrail's "Fully Supported" label.
- The "Conflicting Evidence with No Clear Resolution" label was mapped to "Inconclusive."
- All "I Don't Understand the Claim" labels were excluded from our analysis.[8]

---

[8]"Conflicting Evidence with No Clear Resolution" and "I Don't Understand the Claim" labels were rare: they each represented only 1.6% of all labels.

For 81% of claims, the majority label was used as the final label. For the remaining claims, the author's label was used, for one of three reasons: (a) the claim required evidence beyond the 9-10 articles provided to the Upwork annotators; (b) there was no majority label; or (c) we determined that the annotators had incorrectly applied the instructions or missed relevant evidence. The final label distribution for the 560 claims was 74% "Fully Supported" and 26% "Not Fully Supported."

We assessed alignment between our labels and annotations from the original DiverseSumm dataset. In addition to creating questions for each news story, Huang et al. used the associated articles to generate answers, which were validated by human annotators. They also generated summaries of the articles, and annotators labeled the faithfulness of each summary sentence. We repeated the entailment-based matching procedure described in Appendix B.1 to check whether any of our extracted claims were entailed by either a validated answer or a summary sentence that was deemed faithful. We identified matches for 79 claims (14%), of which 78 were labeled "Fully Supported" in our dataset, indicating near-perfect alignment.

Across the five annotators in our study, Krippendorff's alpha (Krippendorff, 2013; Castro, 2017) was 0.49.[9] For the 56% of claims where no annotators reported uncertainty, alpha was 0.68. For claims where at least one annotator reported uncertainty, alpha dropped to 0.38. These results are comparable to those reported by AmbiFC (Glockner et al., 2024), which we view as the most methodologically similar study: in the condition where at least five annotators labeled claims' entailment with respect to a passage, alpha was 0.55 for the high certainty subset and 0.21 for the low certainty subset (where annotators indicated uncertainty, as in our study). Notably, their passages were capped at 20 sentences, while our annotators worked with much longer texts. The fact that our agreement scores were higher despite the increased task difficulty is encouraging.

We offer two key takeaways based on the inter-annotator agreement results. First, claims where annotators expressed uncertainty had lower levels of agreement, suggesting that asking annotators to indicate uncertainty may help flag challenging or potentially ambiguous cases. Second, after reviewing the claims where annotators disagreed on the correct label, we found that some cases could be attributed to annotation errors, while others reflected reasonable differences in how the claim and/or the evidence were interpreted. This finding suggests that in future studies, it might be helpful to complement single-label annotations with probability distributions reflecting annotator agreement (e.g., as explored in Nie et al., 2020 and Jiang et al., 2023).

### B.2.3 ANNOTATION INSTRUCTIONS

> **Annotation Instructions**
>
> ```
> Thank you so much for participating in this study! We greatly
> appreciate your time and effort. Please read the following
> instructions carefully and let us know if you have any questions.
>
> ## Overview
> You will be provided with one or more folders. Each folder will have
> a code like "Q0", "Q1", "Q2", etc. Each folder corresponds to a
> distinct question or topic area.
>
> Each folder will contain two files:
> 1. claims.xlsx - A spreadsheet where each row corresponds to a single
>   factual "claim," such as "As of 2021, India overtook China as the
> country with the largest population."
> 2. articles.docx - A Word document containing the content of
> approximately 10 news articles.
>
> For each folder, your task is to fact-check every claim in the claims
> .xlsx file using only the information found in the corresponding
> articles.docx file from the same folder.
> ```

[9]For the 13% of claims that required the full article set, the author's labels were omitted from the agreement calculation, since they cannot be fairly compared to the labels from annotators who only had access to the constrained article set.

**Annotation Instructions (Continued)**

You will record your results directly in the claims.xlsx file within each folder. When you're finished, return the same folders you received, with the claims.xlsx file completed in each one. For example, if you were given 10 folders, you should submit those same 10 folders, each containing a completed claims.xlsx file.

## Columns to complete
In the claims.xlsx file, you'll need to complete four columns for each claim: Column C (Label), Column D (Uncertain), Column E (Evidence), and Column F (Comments).

### Column C: Label
Each cell in Column C should have a drop-down menu with the following five options:
1. All parts are supported
2. At least one part is refuted
3. Conflicting evidence with no clear resolution
4. Insufficient evidence (none of the above)
5. I don't understand the claim

Here are some key definitions you'll need in order to correctly apply the labels:
1. A claim is "supported" by the articles if the articles either explicitly state or strongly imply that the claim is true. A simple test here is whether you'd feel comfortable saying: "According to the news articles, <insert claim>."

2. A claim is "refuted" by the articles if the claim is directly contradicted or strongly implied to be false by the articles. A simple test here is whether you'd feel comfortable saying: "Based on the news articles, it's not true that <insert claim>" or "Based on the news articles, it's very unlikely to be true that <insert claim>."
   – IMPORTANT: Lack of evidence does NOT mean the claim is "refuted." If the articles simply don't mention something (e.g., they say nothing about India's or China's population), then that's not refutation – it's "Insufficient evidence" (more on this later).

3. "Conflicting" evidence means that the articles contain both supporting and refuting information. It can also mean that the claim has multiple possible interpretations, and depending on which interpretation you choose, the claim can be supported or refuted. For example, Article 1 says China has the largest population, while Article 2 says India has the largest population. Another example: the claim is "India is larger than China," and Article 1 says India has a bigger population than China, while Article 2 says China is bigger than India in terms of land mass.
   – If you find conflicting evidence, your next step is to determine if: (A) there is a clear resolution to the conflict (e.g., if Article 1 is from 2020 and Article 2 is from 2021, the difference in publication dates might explain the discrepancy, suggesting a possible timeline shift rather than a contradiction), OR (B) there is no clear resolution to the conflict (i.e., both sides appear equally valid and there's no obvious way to determine which is correct).
   – Do NOT resolve conflicts based on which article seems more credible or trustworthy. That is not a valid basis for determining a clear resolution.

**Annotation Instructions (Continued)**

```
4. A claim may have one or more "parts." You can identify these parts
 by asking yourself: "What must be true in order for the entire claim
  to be true?" In the example claim above - "As of 2021, India
overtook China as the country with the largest population" - there
are three things that must be true in order for the entire claim to
be true: (1) India became the country with the largest population,
(2) this happened as of 2021, and (3) prior to 2021, China had the
largest population.

Now, here are the rules for labels:
1. If all parts of the claim are supported by the articles, then you
should select label 1: "All parts are supported."
2. If at least one part of the claim is refuted by the articles, then
 you should select label 2: "At least one part is refuted".
   - For example, even if the articles confirm that India now has the
     largest population (part 1) and that China used to have the
   largest population (part 3), if they indicate that India overtook
   China in 2022, not 2021 as claimed, then part 2 is refuted, so you
     would select the "At least one part is refuted" label.
3. If there is conflicting evidence that does not have a clear
resolution for at least one part of the claim, then you should select
 label 3: "Conflicting evidence with no clear resolution."
   - If you feel the conflicting evidence does have a clear
   resolution, then you should just treat that evidence as supportive
    or refuting, whichever resolution is preferred.
4. If none of the above options apply, then you should select label
4: "Insufficient evidence (none of the above)". This covers cases
where: (1) no parts of the claim are refuted by the articles (or else
 we would've selected label 2) AND (2) no parts of the claim have
conflicting evidence that can't be resolved (or else we
would've selected label 3) AND (3) at least one part of the claim is
missing support.
   - For example, if the articles indicate that India surpassed China
     as the most populous country, but there are no dates provided,
   then the claim is not refuted and there's no conflicting evidence,
    but it's also not fully supported, so you would select the "
   Insufficient evidence (none of the above)" label.
5. If you don't understand the claim for whatever reason, then you
should select label 5: "I don't understand the claim."

You should follow the order of priority described above when
assigning labels. For example, if one part of the claim is refuted by
 the articles (label 2), but another part has conflicting evidence
with no resolution (label 3), then you should assign label 2.

IMPORTANT: Do NOT use any personal knowledge or outside sources in
the fact-checking process. For example, if the claim is "The capital
of the United States is Washington, D.C.," but the articles don't
contain any evidence that supports or refutes this claim, it doesn't
matter that you know the claim is true. You must still select the "
Insufficient evidence (none of the above)" label.

### Column D: Uncertain
Fact-checking is often subjective - there won't always be a clear-cut
 "correct" answer. If you feel uncertain about the label you selected
 for any reason, please enter TRUE in Column D.

You will NOT be penalized for indicating uncertainty - in fact,
it's valuable for our analysis, as it helps identify potentially "
tricky" cases that are ambiguous or open to interpretation. If
```

**Annotation Instructions (Continued)**

```
you're confident in your label, you can simply leave Column D blank
or enter FALSE.

Potential reasons you might feel uncertain include:
- You found some relevant evidence, but you're unsure whether it's
strong enough to support or refute the claim.
- You're not sure whether you interpreted the claim correctly.

### Column E: Evidence
If you selected one of the following labels, then you must provide at
 least one direct quote from the articles to justify your choice:
1. "All parts are supported" - Provide evidence that supports all
parts of the claim
2. "At least one part is refuted" - Provide evidence that directly
contradicts the claim or strongly implies that it's false
3. "Conflicting evidence with no clear resolution" - Provide evidence
 that both supports and refutes the claim

Your evidence does NOT need to be exhaustive. For example, if there
are 10 quotes that say India's population is larger than China's, it'
s sufficient to provide one of them. However, if the claim has
multiple parts and you selected the label "All parts are supported,"
make sure that the evidence you include (whether it's a single quote
or multiple) collectively supports all parts.
Do not include commentary or explanations in this column - only paste
 the relevant quote(s). Save any comments for Column F.

### Column F: Comments
This column is optional but highly recommended - especially if you
felt uncertain or selected the "Conflicting evidence with no clear
resolution" label. Use Column F to:
- Briefly explain your reasoning behind the label
- Clarify how you interpreted the claim
- Note anything unusual or borderline about the evidence

These comments help us review your work and understand your decision-
making process.

## Process & Tips
Here's the recommended process to follow for each folder:
1. Open articles.docx and skim through the content to get familiar
with the topic area.
2. Read the first claim carefully. Make sure you fully understand the
 claim. If it's unclear or confusing for any reason, select the label
 "I don't understand the claim." Also consider whether the claim has
multiple possible interpretations that you need to consider.
3. Break down the claim into parts (or determine that there's only
one part). Ask yourself: "What must be true in order for the entire
claim to be true?"
4. Search for evidence in the articles. Look for sentences that
support or refute any part of the claim.
   - Tip: Use the "Find" function (Ctrl+F on Windows, Command+F on
   Mac) to search for key terms.
   - If you've searched thoroughly and found no relevant evidence,
   select the "Insufficient evidence (none of the above)" label.
   - If you've found evidence that seems to be conflicting
   (i.e., there is both evidence that supports at least part of the
   claim as well as evidence that refutes it), determine whether you
   think there's a clear resolution to the conflict, or not.
5. Fill out Columns C, D, E, and F, using the guidance above.
6. Move on to the next claim!
```

**Annotation Instructions (Continued)**

– Tip: Some claims may overlap. For example, consider Claim 1 = " Jill is John's manager" and Claim 2 = "John asked his manager Jill for a promotion." If you find that Claim 1 is refuted, you already know that a part of Claim 2 is also false. You can reuse the same evidence when labeling Claim 2.

## Additional Clarifications

When should you use the label "Conflicting evidence with no clear resolution"?

This label should only be used when you have both evidence that supports the claim as well as evidence that refutes the claim, and you're not sure which is correct. It can also be used if you feel there are multiple possible interpretations of the claim, one of which is supported by the evidence, another one of which is not, and it's not clear which interpretation/resolution is correct.

If there's Insufficient evidence that feels relevant, or only very weakly relevant, you should pick "Insufficient evidence."

If there is some relevant evidence and you're just not sure whether it's "strong" enough to count as supporting (or refuting) the claim, then you should decide whether you: (A) lean towards it being strong enough to count as supporting (or refuting), so pick "all parts are supported" (or "at least one part is refuted"), or (B) lean towards it NOT being strong enough to count as supporting (or refuting), so pick "Insufficient evidence". In both cases, it would be good to set "Uncertain" to True (don't be afraid to do this!). But, to reiterate, the "conflicting" label is for when you feel it's roughly equally likely that the claim is supported or refuted, NOT when there's only one real option for the label (e.g., supported) and you're just not sure if the evidence is strong enough to warrant that label.

## C VERITRAIL DETAILS

### C.1 ALGORITHM

Algorithm 1 details VeriTrail's verification procedure for a single claim $c$. The procedure assumes access to a DAG $G = (V, E)$ representing a generative process (see §2), including the terminal node $v^*$, root nodes $V_0$, and the source function src.

---

**Algorithm 1** VeriTrail

---

**Require:** claim $c$, max NotFullySupported iterations $q$

1: $evidence\_trail \leftarrow [\,]$
2: $consec\_not\_supp \leftarrow 0$
3: $checked \leftarrow \emptyset$
4: $roots\_with\_ev \leftarrow \emptyset$
5: $all\_verdicts \leftarrow [\,]$
6: $nodes\_to\_check \leftarrow \text{src}(v^*)$

7: **while** true **do**
8:     $(evidence,\ nodes\_with\_ev) \leftarrow \text{get\_evidence}(c,\ nodes\_to\_check)$
9:     **if** $evidence = \emptyset$ **then**
10:         $verdict \leftarrow \text{NotFullySupported}$
11:     **else**
12:         add $evidence$ to $evidence\_trail$
13:         $roots\_with\_ev \leftarrow roots\_with\_ev\ \cup\ \big(nodes\_with\_ev \cap V_0\big)$
14:         $verdict \leftarrow \text{get\_verdict}(c,\ evidence,\ nodes\_with\_ev)$
15:     **end if**
16:     add $verdict$ to $all\_verdicts$
17:     $checked \leftarrow checked\ \cup\ nodes\_to\_check$
18:     **if** $verdict = \text{NotFullySupported}$ **then**
19:         $consec\_not\_supp \leftarrow consec\_not\_supp + 1$
20:         $nodes\_to\_check \leftarrow \bigcup_{n \in nodes\_to\_check} \text{src}(n)$
21:     **else**
22:         $consec\_not\_supp \leftarrow 0$
23:         $nodes\_to\_check \leftarrow \bigcup_{n \in nodes\_with\_ev} \text{src}(n)$
24:     **end if**
25:     $nodes\_to\_check \leftarrow (nodes\_to\_check \setminus checked)\ \cup\ roots\_with\_ev$
26:     **if** $nodes\_to\_check = roots\_with\_ev$ **then**
27:         **break**
28:     **end if**
29:     **if** $nodes\_to\_check = \emptyset$ **then**
30:         $verdict \leftarrow \text{NotFullySupported}$
31:         **break**
32:     **end if**
33:     **if** $consec\_not\_supp = q$ **then**
34:         **break**
35:     **end if**
36: **end while**

37: **return** $(verdict,\ evidence\_trail,\ all\_verdicts)$

---

## C.2 PROMPTS

### C.2.1 EVIDENCE SELECTION PROMPT

---

**Evidence Selection System Prompt**

You are an extremely smart, thorough, and meticulous assistant. You will be given a collection of excerpts from one or more sources. Each excerpt is preceded by a label like [[1]], and each sentence in the excerpts has an ID. You will also be given a question of the form "Is there any information in the excerpts that indicates <proposition>?" Your task is to answer the question.

Note the following rules:
- Sometimes the proposition can be further decomposed into sub-propositions. For example, if the proposition is "There have been advancements in clean energy and desalination technologies," the sub-propositions are: "There have been advancements in clean energy" and "There have been advancements in desalination technologies." If information in the excerpts strongly implies the truth or falsehood of at least one sub-proposition, it should be included in your answer.
- You will only include information that STRONGLY implies a sub-proposition's truth or falsehood. You will NOT include weak implications. If you are not sure whether a sub-proposition is a STRONG or WEAK implication, you should defer towards including it in your answer.
- You will put yourself in the shoes of a careful reader who interprets the text holistically, considering both explicit statements and implied meaning. For example, if the claim is "John emphasizes the importance of mentorship programs", and John never explicitly says that mentorship programs are important but it's clear that he values them because he speaks of his attempts to establish mentorship programs and he comes across as passionate about them, then a careful reader would find that the proposition is strongly implied.
- If the proposition is something like "John found X", "John reported X", "John emphasizes X", etc. (where John can be replaced with any entity or entities), it should be interpreted as a statement about what John says or does. For example, if the proposition is "John highlights that transparent communication is a critical part of Project Alpha", and the excerpts indicate that transparent communication is a critical part of Project Alpha, but they are missing the critical context that this is something John highlights, then they would NOT strongly imply the truth or falsehood of the proposition. Let's call this the Statements and Actions Rule.
- You will NOT use any external knowledge beyond what is stated in the provided excerpts.
- It is EXTREMELY important that you cite the correct IDs. You will be heavily penalized if you attribute information to the wrong ID.

Your output must adhere to the following format exactly.
# Question: <insert full question>
# Proposition: <insert proposition>

## Step 1: Decompose proposition into sub-propositions that cannot be further decomposed (two rounds)
<Decompose the proposition (P) into a list of independent sub-propositions SP = [SP1, SP2, ...]. If the proposition cannot be decomposed into multiple independent sub-propositions, return a single-label list. Make sure to follow the Statements and Actions Rule. Ensure that the SP do not contain any unverifiable components (

---

**Evidence Selection System Prompt (Continued)**

```
e.g., "extensive", "significant", "substantial", etc.) from P. You
will do this in two rounds, to ensure that the sub-propositions
cannot be decomposed any further. For example:
P = "As the CEO of Company X, John's frequent emphasis on the
importance of solar and wind energy has contributed to their
mainstream acceptance."
Round 1: SP without unverifiable components = [
"John is the CEO of Company X",
"John has emphasized the importance of solar and wind energy",
"John's emphasis on the importance of solar and wind energy has
contributed to their mainstream acceptance"
]
Round 2: SP without unverifiable components = [
"John is the CEO of Company X",
"John has emphasized the importance of solar energy",
"John has emphasized the importance of wind energy",
"John's emphasis on the importance of solar energy has contributed to
 its mainstream acceptance",
"John's emphasis on the importance of wind energy has contributed to
its mainstream acceptance"
]>

## Step 2: Provide an overview of sentences
<Provide an accurate overview of the sentences in the excerpts with
respect to the question, without adding any interpretations or making
 any assumptions. The overview should be fully entailed by the
excerpts. For example, if the question asks whether there have been
advancements in clean energy and a sentence says there is a potential
 for advancements in clean energy, the overview will NOT say "
mentions advancements in clean energy" as this misrepresents the
sentence; it will say "mentions a potential for advancements in clean
 energy". It can be very helpful to organize the sentences by excerpt
. Provide a point for each sentence WITHOUT quoting it. If there aren
't any relevant sentences, state "NO RELEVANT SENTENCES" and
terminate your output here. It is EXTREMELY important that you do not
 overlook any relevant sentences.>

## Step 3: Test each sentence or each range of sentences
<For each sentence or range of sentences you identified in Step 2,
print the sentence ID or range of sentence IDs then complete ALL of
the bulleted statements below. If it's not possible to make a good
faith completion for a statement (i.e., you should NOT claim that the
 sentence states something when it does not, or that it fails to
state something when it does), you should put "N/A" for that
statement. Remember that you are NOT allowed to use any information
outside of the provided excerpts. You MUST cover ALL of the sentences
 or ranges of sentences you identified in Step 2.
- SP = <insert the SP from Step 1 that is most relevant to the
sentence or range of sentences>
- One might use the following quote to argue that the sentence(s)
strongly implies (NOT necessarily explicitly states) the truth or
falsehood of SP: "..."
- One might use the following quote(s) from the remaining sentence ID
(s) in the excerpts as additional context: "..." or "N/A"
- A careful reader trained to look for STRONG IMPLICATIONS, which is
a weaker standard than explicit statements, and to consider the
sentence(s) holistically would reason as follows: <insert step-by-
step reasoning, then clearly state the conclusion about whether or
not it could be interpreted as a strong implication; remember that if
 you're not sure whether it's a strong implication, you should defer
towards including it>.>
```

**Evidence Selection System Prompt (Continued)**

```
## Step 4: Final submission
<Insert EITHER (1) "The excerpts do not contain any information that
strongly implies any sub-proposition" OR (2) "The following sentences
 provide a strong implication: [<insert ALL sentence IDs where strong
 implication is the conclusion from Step 3; do NOT include any
excerpt labels, e.g., [[1]]:5 is incorrect vs. 5 is correct; ranges
are allowed for consecutive sentence IDs, e.g., 5-10>] with the
following sentence(s) providing essential context: [<insert ALL
sentence IDs needed as context for the sentence IDs that provide a
strong implication; if no context is needed because the sentence IDs
independently provide strong implication, leave this empty>] Here is
a complete summary covering ALL information in the sentence(s) that
is relevant to at least one sub-proposition and ALL context necessary
 to understand them and their connection to the sub-proposition(s),
without mentioning what is implied or indicated: <insert an accurate
description of the information contained in the sentence(s) and their
 connection to the sub-proposition(s); always use full names for
entities when they are provided; do NOT just quote the sentences; do
not speculate about what is implied or indicated.> Here are some
comments on what is missing or unclear: <insert here, or "N/A">
```

**Evidence Selection User Prompt**

```
Excerpts:
{excerpts}

Question:
{question}

Example sub-propositions (SP) that may need to be decomposed further:
{sub_claims}
```

## C.2.2   VERDICT GENERATION PROMPT

**Verdict Generation System Prompt**

```
You are an extremely smart, thorough, and meticulous assistant. You
will be given a collection of excerpts from one or more sources. Each
 excerpt is preceded by a label like [[1]], and each sentence in the
excerpts has an ID. You will also be given a
claim. Your task is to answer the following question: Do the excerpts
 justify the entire claim?

In order for the excerpts to justify the entire claim, the excerpts
must STRONGLY imply that the entire claim is true. This means that a
careful reader of the excerpts would naturally infer the entire claim
 without needing to make any assumptions or access any external
information. Note that strong implication is a weaker standard than
explicit statement. Also note that WEAK implication is NOT sufficient
. For example, if the claim is "John highlights the importance of
collaboration in driving innovation" and the only relevant evidence
in the excerpts is that John worked on several team projects, the
excerpts would NOT justify the entire claim.

There are 4 possible cases where the excerpts do NOT justify the
entire claim:
1. The excerpts contradict at least one part of the claim
```

**Verdict Generation System Prompt (Continued)**

2. The excerpts strongly imply that at least one part of the claim is false
3. At least one part of the claim is only weakly implied by the excerpts
4. At least one part of the claim is not addressed by the excerpts

Note the following rules:
- The claim is extracted from an answer to a question about a collection of documents. Therefore, if the claim is something like "X is mentioned" or "X is discussed," it should be interpreted as a statement about what is mentioned or discussed in the documents.
- If the claim is something like "John found X", "John reported X", "John emphasizes X", etc. (where John can be replaced with any entity or entities), it should be interpreted as a statement about what John says or does. For example, if the claim is "John highlights that transparent communication is a critical part of Project Alpha", and the excerpts indicate that transparent communication is a critical part of Project Alpha, but they are missing the critical context that this is something John highlights, then they would NOT justify the entire claim. Let's call this the Statements and Actions Rule.
- You will NOT use any external knowledge beyond what is stated in the provided excerpts.
- You will put yourself in the shoes of a careful reader who interprets the text holistically, considering both explicit statements and implied meaning. For example, if the claim is "John emphasizes the importance of mentorship programs", and John never explicitly says in the text that mentorship programs are important but it's clear that he values them because he speaks of his attempts to establish mentorship programs and he comes across as passionate about them, then a careful reader would find that the excerpts justify the entire claim.
- You will operate under the assumption that the excerpts contain all information required to make a determination. For example, if the claim is "John led three teams" and the excerpts are from an interview where John only mentions one team that he led, you will NOT argue that the excerpts do not provide a comprehensive list of all teams that John led so a determination cannot be made. Instead, you will consider the excerpts to be the only source of truth and since they only support the conclusion that John led one team, the excerpts do NOT justify the entire claim. Similarly, if one source in the excerpts provides a list of teams and another source indicates that some teams were led by John, it IS valid to cross-reference the lists to determine the number of teams John led.

Your output must adhere to the following format exactly. Do NOT remove the instructions.
1: Claim = <insert claim>

2: Does the Claim have multiple possible interpretations? If yes, specify them, then clearly state which one you believe most people would agree with - you will use this interpretation for the rest of your output. If there are distinct aspects of the Claim that must be true for the Claim to be true, enumerate them (e.g., "John worked at (1) Company A and (2) Company B"). Also identify any unverifiable components of the Claim (e.g., "extensive", "significant", "substantial", etc.) Print "ClarifiedClaim = <insert clarified version of the Claim>".

3: Quote the relevant sentences in the text with respect to the ClarifiedClaim without any interpretations or judgments, making sure

## Verdict Generation System Prompt (Continued)

```
to include the sentence IDs. Do NOT cover sentences about the lack of
  information, e.g., "there is no explicit mention of X". If there
aren't any relevant sentences, state "NO RELEVANT SENTENCES" and
terminate your output here. If there are likely more than 10 relevant
  sentences, pick the 10 most important ones.
<insert stream of consciousness thought process; use bullet points or
  numbered lists if needed>

4: Identify ALL pieces of evidence from step 3 that are CONFLICTING (
i.e., one piece of evidence indicates X is true while another
indicates X is false), outline the possible resolutions, and
determine whether or not the excerpts STRONGLY imply that one
resolution is preferred over the other(s). If yes, clearly state
which one is preferred, and use this information in your final
deliberation. If not, you will DISCARD this issue in your final
deliberation (i.e., you will treat it as if the resolution is unknown
, so it cannot be used to make a determination). Make sure to include
  the sentence IDs in your output.

5: Identify ALL pieces of evidence from step 3 that are DEBATABLE (i.
e., people could reasonably disagree on what the evidence
means, what it implies with respect to the ClarifiedClaim, and/or the
  strength of the implication), outline the possible conflicting
positions, and determine whether or not one position is more
compelling than the other(s). If yes, clearly state which one is more
  compelling, and use this information in your final deliberation. If
not, you will DISCARD this issue in your final deliberation (i.e.,
you will treat it as if the resolution is unknown, so it cannot be
used to make a determination). Make sure to include the sentence IDs
in your output.

6: List ALL sentence IDs from step 3 that were NOT included in steps
4 and 5, then quote them. These pieces of evidence are CLEAR in their
  meaning and implication for the ClarifiedClaim.

7: Given your analysis of the evidence in steps 4-6, and considering
that there may be parts of the ClarifiedClaim that are NOT addressed
by the evidence, does the NON-DISCARDED evidence from the excerpts
justify (i.e., STRONGLY imply) the ENTIRE claim? Remember that strong
  implication is a weaker standard than explicit statement, but weak
implication and speculations are NOT sufficient. First, walk through
your reasoning step-by-step; do NOT jump straight to the conclusion.
Then print, "I submit the following answer: <insert 'Excerpts justify
  the entire ClarifiedClaim' or 'Excerpts do not justify the entire
ClarifiedClaim' or 'Cannot determine if Excerpts justify the entire
ClarifiedClaim'>. Only use 'Cannot determine if Excerpts justify the
entire ClarifiedClaim' if all evidence was DISCARDED.
```

## Verdict Generation User Prompt

```
Excerpts:
{excerpts}

Claim:
{claim}
```

The output "Excerpts justify the entire ClarifiedClaim" corresponds to VeriTrail's "Fully Supported" verdict. The output "Excerpts do not justify the entire ClarifiedClaim" corresponds to the "Not Fully Supported" verdict. The output "Cannot Determine if Excerpts justify the entire ClarifiedClaim" corresponds to the "Inconclusive" verdict.

# D    COMPUTATIONAL COST

Table 2 compares the average cost per claim for VeriTrail, all baseline methods except the NLI models, and human annotation.[10] Any retries due to invalid outputs were also factored into the estimates. We note that RAG consists of two parts: (1) a one-time embedding of all source text chunks, and (2) per-claim processing (embedding the claim, retrieving $k$ chunks, and generating a verdict). We compute the *amortized* per-claim cost, which divides the one-time embedding cost across all claims and adds the result to the average per-claim processing cost. For completeness, the one-time embedding cost for RAG was $0.17 for DiverseSumm+ and $0.02 for FABLES+.

**Table 2:** Average cost per claim ($) for the FABLES+ (F) and DiverseSumm+ (D) datasets. VeriTrail's $q$ hyperparameter specifies the number of consecutive "Not Fully Supported" verdicts that will trigger termination of the hallucination detection process. RAG's $k$ hyperparameter specifies the number of top-ranked chunks that were retrieved.

| Method | $/Claim | |
|---|---|---|
| | **F** | **D** |
| VeriTrail (`DeepSeek-V3`, $q = 1$) | 0.06 | 0.12 |
| VeriTrail (`gemini-2.5-flash-preview-04-17`, $q = 1$) | 0.09 | 0.14 |
| VeriTrail (`mistral-large-2411`, $q = 1$) | 0.46 | 0.83 |
| VeriTrail (`gpt-4o-2024-0806`, $q = 1$) | 0.90 | 1.22 |
| VeriTrail (`gpt-4o-2024-0806`, $q = 2$) | 1.15 | 1.61 |
| VeriTrail (`gpt-4o-2024-0806`, $q = 3$) | 1.20 | 2.12 |
| RAG (`gpt-4o-2024-0806`, $k = 5$) | 0.06 | 0.03 |
| RAG (`gpt-4o-2024-0806`, $k = 15$) | 0.12 | 0.04 |
| RAG (`gpt-4o-2024-0806`, $k = 25$) | 0.21 | 0.06 |
| GPT-4.1 Mini | 0.06 | 0.54 |
| Gemini 1.5 Pro | 0.38 | 3.37 |
| Human Annotation | 1.65 | 2.41 |

The cost estimates in Table 2, combined with the performance results in Table 8 in Appendix E.7, demonstrate that VeriTrail can achieve strong performance at low cost. For example, using Gemini-2.5-Flash with $q = 1$, VeriTrail costs only $0.09 - $0.14 per claim, while still outperforming the baselines. These results are particularly noteworthy given VeriTrail's significantly larger verification burden. Unlike the baseline methods, which only provide a faithfulness verdict by comparing claims directly to the source material, VeriTrail also provides traceability by constructing an evidence trail through intermediate outputs. For example, in DiverseSumm+, the baselines only evaluate ~3K root nodes, whereas VeriTrail must evaluate an additional 110K intermediate nodes.

In terms of time complexity, the worst-case for VeriTrail is $\mathcal{O}(|V|)$ per claim, where $|V|$ is the number of nodes in the input DAG. However, this upper bound assumes that the claim must be verified against all nodes. In practice, considerably fewer nodes are evaluated. For example, when $q = 3$, the average percentage of nodes verified was 47% for FABLES+ and 1% for DiverseSumm+; when $q = 1$, the corresponding averages were 38% and 0.3%, respectively.

VeriTrail's efficiency stems from the following aspects of its design:

1. **Early Termination.** If a claim reaches $q$ consecutive "Not Fully Supported" verdicts, the verification process terminates, even if the root nodes have not yet been reached. Therefore, lower $q$ values allow the process to terminate earlier, reducing computational cost. Table 2 confirms that lower $q$ values are associated with lower average cost per claim.

2. **Selective Verification.** As described in §3.1.4, after a "Fully Supported" or "Inconclusive" verdict, only the source nodes of nodes that yielded evidence are verified – not all possible

---

[10]For DiverseSumm+, our cost estimate for human annotation is the total amount spent on the annotation study ($1,350) divided by the number of claims in the dataset (560). For FABLES+, we used the cost reported in the original FABLES paper ($5,200) divided by the number of claims in their dataset (3,158). We do not include the NLI models as their costs vary depending on the type of GPU used.

source nodes. This approach avoids wasting computation on nodes that are unlikely to contain relevant information.

3. **Reverse Traversal.** VeriTrail verifies claims in the reverse order of the generative process: it starts from the source nodes of the terminal node $v^*$ and proceeds towards the root nodes $V_0$. Because selective verification progressively narrows the search space and low $q$ values enable early termination, VeriTrail tends to verify a larger proportion of nodes in later stages (closer to $v^*$) than in earlier stages (closer to $V_0$). Since earlier-stage nodes are typically larger (e.g., a book chapter is larger than a chapter summary), verifying fewer of them reduces cost.

# E   ADDITIONAL EXPERIMENTS

All experiments in this section were conducted on a random subset of the data: 6 books from FABLES+ and 7 questions from DiverseSumm+. This subset includes 31% of claims from FABLES+ and 34% from DiverseSumm+, totaling 415 claims. Unless otherwise noted, VeriTrail and RAG results in this section were produced using `gpt-4o-2024-08-06`, with $q = 1$ for VeriTrail.

## E.1   UNDERSTANDING VERITRAIL'S PERFORMANCE

In this section, we analyze which factors contributed to VeriTrail's performance gains. As noted in §4.3, all LM-based methods used the same verdict prompt, so VeriTrail's Verdict Generation step cannot explain why it outperformed the baseline methods. Instead, VeriTrail differs from the baselines in two key ways: (1) it traces claims through intermediate outputs rather than checking them directly against the source material, and (2) it uses LM-based Evidence Selection prior to Verdict Generation.

To isolate the contribution of each component, we created a variant of VeriTrail that retained its tracing mechanism and Verdict Generation prompt but replaced its LM-based Evidence Selection step with the embedding-based retrieval approach used in RAG, the best-performing baseline. At each iteration, this variant retrieved the top-$k$ nodes most similar to the claim and used them as input for Verdict Generation. We tested $k \in \{5, 15, 25\}$. If the verdict was "Not Fully Supported," verification terminated (as in the original VeriTrail with $q = 1$); otherwise, the source nodes of the retrieved nodes were evaluated in the next iteration.

We refer to this variant as **VT-RAG** (VeriTrail-RAG hybrid), and compare it to the original VeriTrail with $q = 1$ (**VT**) and **RAG**. These methods can be summarized as follows:

- VT = tracing + LM-based Evidence Selection
- VT-RAG = tracing + embedding-based Evidence Selection
- RAG = no tracing + embedding-based Evidence Selection

If tracing through intermediate outputs is the primary driver of VeriTrail's performance gains, we expect VT ≈ VT-RAG > RAG. If LM-based Evidence Selection is the primary driver, we expect VT > VT-RAG and VT > RAG. If both factors contribute, we expect VT > VT-RAG > RAG.

Results are shown in Table 3. Across both datasets, the original VeriTrail (VT) outperformed the other methods. For FABLES+, we observe VT > RAG > VT-RAG, suggesting that VeriTrail's Evidence Selection step is the main reason it outperforms the other methods. For DiverseSumm+, we observe VT > VT-RAG > RAG, indicating that both the Evidence Selection step and tracing through intermediate outputs contribute to VeriTrail's performance gains.

**Table 3:** Hard prediction results (%) on the FABLES+ (F) and DiverseSumm+ (D) datasets, comparing standard VeriTrail with $q$=1 (VT), RAG, and a hybrid method (VT-RAG). For RAG and VT-RAG, $k$ denotes the number of top-ranked chunks that were retrieved. We report macro $F_1$, balanced accuracy (Bal. Acc.), and class-specific precision and recall for fully supported (FS) and not fully supported (NFS) claims. Bolded values indicate the highest score per column.

| Method | $k$ | Macro $F_1$ | | Bal. Acc. | | Precision$_{FS}$ | | Recall$_{FS}$ | | Precision$_{NFS}$ | | Recall$_{NFS}$ | |
|--------|-----|------|------|------|------|------|------|------|------|------|------|------|------|
| | | F | D | F | D | F | D | F | D | F | D | F | D |
| VT | - | **69.2** | **81.2** | **80.5** | **85.8** | 96.6 | 95.7 | 77.2 | 83.0 | **38.2** | 62.9 | 83.9 | 88.6 |
| RAG | 5 | 58.9 | 71.5 | 66.5 | 71.0 | 91.7 | 85.5 | 71.7 | 87.4 | 26.8 | 58.5 | 61.3 | 54.5 |
| VT-RAG | 5 | 59.6 | 75.5 | 75.1 | 82.5 | 96.7 | **96.2** | 63.0 | 74.1 | 28.4 | 53.3 | 87.1 | **90.9** |
| RAG | 15 | 66.6 | 72.2 | 72.7 | 70.2 | 93.1 | 84.5 | **81.0** | 92.6 | 36.4 | **67.7** | 64.5 | 47.7 |
| VT-RAG | 15 | 61.5 | 77.7 | 75.4 | 82.8 | 96.1 | 94.7 | 66.8 | 79.3 | 29.9 | 57.6 | 83.9 | 86.4 |
| RAG | 25 | 63.9 | 69.3 | 70.0 | 67.1 | 92.4 | 82.9 | 78.8 | **93.3** | 32.8 | 66.7 | 61.3 | 40.9 |
| VT-RAG | 25 | 51.9 | 73.8 | 70.4 | 78.3 | **96.9** | 92.0 | 50.5 | 77.0 | 23.5 | 53.0 | **90.3** | 79.5 |

### E.2 VeriTrail Input Size Limit

As explained in §3.1.2 and §3.1.3, VeriTrail allows users to set an input size limit per prompt for both Evidence Selection and Verdict Generation. If no limit is specified, the LM's context window size is used. Separate limits can be specified for root and non-root nodes.

In our experiments, we set the Evidence Selection limit to 40 sentences per prompt for all nodes. For Verdict Generation, we set the limit to 200 sentences for non-root nodes, with no limit for root nodes. This means that for Evidence Selection, nodes were split into sentences and divided into prompts of up to 40 sentences each. For Verdict Generation, the prompt was capped at 200 sentences if no root nodes were included in the evidence; if any root nodes were included, the input limit defaulted to the context window size.

We evaluated the effect of the input size limit through an ablation study. We did not vary the limit for Verdict Generation because all evidence must fit in a single prompt. Decreasing the limit would only force compression or removal of evidence, which is unlikely to improve performance. Increasing the limit would have minimal effect: for non-root nodes, our default limit was rarely exceeded, and for root nodes, the limit was already set to the full context window size. Instead, we focused our ablation study on input size limits for Evidence Selection. Unlike Verdict Generation, Evidence Selection allows any number of prompts, making it a more meaningful setting for studying the impact of input size limits.

We hypothesized that the key risk of a lower limit (i.e., using many short prompts) is context loss. For example, consider a node containing the following sentences: "*John began his career at Company A. He later worked at Company B.*" If both sentences appeared in the same Evidence Selection prompt, they would likely be selected as evidence for the claim "*John worked at Company A prior to Company B.*" However, if the sentences were split across different prompts, the second sentence might not be selected because it would be unclear who "*He*" refers to without the preceding sentence.

Conversely, we hypothesized that the key risk of a higher limit (i.e., using a few long prompts) is reduced recall. LMs are known to struggle with needle-in-a-haystack retrieval, where they must identify specific pieces of information within a long context (Kamradt, 2023). Evidence Selection is even more challenging than traditional needle-in-a-haystack tasks because (a) multiple relevant sentences ("needles") may exist, and (b) complex reasoning is sometimes required to assess the logical relationship between each sentence and the claim.

We evaluated four input size limits in addition to our default of 40 sentences per prompt: 20, 80, 160, and 320 sentences. Table 4 shows the results. On DiverseSumm+, the default setting achieved the highest macro $F_1$ and balanced accuracy. On FABLES+, the 160-sentence condition performed best.

**Table 4:** Effect of input size limits (in sentences) for Evidence Selection on hard prediction performance for FABLES+ (F) and DiverseSumm+ (D). We report macro $F_1$, balanced accuracy (Bal. Acc.), and class-specific precision and recall for fully supported (FS) and not fully supported (NFS) claims as percentages. Bolded values represent the highest score in each column.

| Input Limit | Macro $F_1$ | | Bal. Acc. | | Precision$_{FS}$ | | Recall$_{FS}$ | | Precision$_{NFS}$ | | Recall$_{NFS}$ | |
|---|---|---|---|---|---|---|---|---|---|---|---|---|
| | F | D | F | D | F | D | F | D | F | D | F | D |
| 20 | 66.2 | 80.3 | 81.1 | 84.6 | 97.9 | 95.0 | 71.9 | **81.4** | 34.1 | 62.3 | 90.3 | 87.8 |
| 40 | 67.4 | **81.0** | 78.6 | **85.6** | 96.1 | 95.8 | **76.6** | **81.4** | 35.7 | **62.9** | 80.6 | 89.8 |
| 80 | 65.4 | 72.4 | 78.4 | 76.0 | 96.6 | 89.9 | 72.9 | 76.4 | 33.3 | 52.9 | 83.9 | 75.5 |
| 160 | **67.8** | 76.1 | **82.1** | 82.7 | 97.9 | **96.3** | 74.0 | 73.6 | **35.9** | 54.9 | 90.3 | **91.8** |
| 320 | 65.4 | 75.4 | 81.7 | 81.7 | **98.5** | 95.4 | 69.8 | 73.6 | 33.3 | 54.3 | **93.5** | 89.8 |

One possible explanation for this difference is the nature of the source material. Claims from DiverseSumm+ were evaluated against many short, self-contained articles, while claims from FABLES+ were evaluated against a single long book composed of interdependent parts. Articles also tend to be written in a compact, expository style, whereas fiction books are typically more verbose, with ideas unfolding gradually through narration and dialogue. As a result, evidence required to verify FABLES+ claims is less likely to be concentrated within a narrow span of text, and therefore, may

benefit more from context preservation. However, in both datasets, performance declined at the highest limit we tested (320 sentences), suggesting that – regardless of source material type – there may be a tipping point where the benefits of context preservation are outweighed by losses in recall.

### E.3 SOFT PREDICTION

In the soft prediction setting, introduced in §4.3, methods produce a continuous score representing the probability that a claim is fully supported. Only AlignScore, INFUSE, and Llama-3.1-Bespoke-MiniCheck-7B natively produce continuous scores. For the remaining methods (except Gemini 1.5 Pro, which was excluded due to cost constraints), we sampled three verdicts at a temperature of 0.2 and calculated the proportion labeled "Fully Supported." This approach is predicated on prior work demonstrating that the consistency of an LM's outputs across samples can be used as a proxy for confidence (Wang et al., 2023; Tian et al., 2023).

Recall that VeriTrail (a) performs Evidence Selection before Verdict Generation, and (b) generates interim verdicts as it traverses the DAG before producing a final verdict. To approximate different confidence thresholds for these intermediate steps, we tested three thresholds, $t \in \{1, 2, 3\}$:

- For each setting of $t$, during Evidence Selection, we generated three outputs using a temperature of 0.2. A sentence was included as evidence only if it was selected in at least $t$ runs. If no sentence met this condition for a given claim, verification was terminated. As no final verdict was generated in these terminated cases (10% and 11% of claims for FABLES+ and DiverseSumm+, respectively), we excluded them from soft prediction evaluation across all methods.

- For Verdict Generation, we likewise generated three outputs using a temperature of 0.2. For each interim iteration, a verdict "passed" if it appeared in at least $t$ outputs. If multiple verdicts passed, we selected the one that appeared most often across the outputs. If no verdicts passed, or passing verdicts tied in frequency, the verdict for that iteration was set to "Inconclusive."

- We used the proportion of "Fully Supported" verdicts from the final iteration as the soft prediction score.

We evaluated soft predictions using Area Under the ROC Curve (AUROC), which measures a method's ability to distinguish supported from unsupported claims across varying classification thresholds. Table 5 shows the results for both datasets. All VeriTrail variants outperformed the baseline methods, with the $t = 2$ variant achieving the best results.

**Table 5:** Soft prediction results for the FABLES+ (F) and DiverseSumm+ (D) datasets. For RAG and AlignScore, we report the best-performing configuration by macro $F_1$: RAG uses $k = 20$ for FABLES+ and $k = 15$ for DiverseSumm+; Llama-3.1-Bespoke-MiniCheck-7B (denoted "Bespoke-MiniCheck-7B") uses $k = 4$ for FABLES+ and $k = 1$ for DiverseSumm+; AlignScore uses $k = 1$ for FABLES+ and $k = 2$ for DiverseSumm+. Bolded values represent the highest AUROC in each column.

| Method | AUROC | |
|---|---|---|
| | **F** | **D** |
| VeriTrail ($t = 1$) | 0.86 | 0.79 |
| VeriTrail ($t = 2$) | **0.88** | **0.87** |
| VeriTrail ($t = 3$) | 0.85 | 0.80 |
| RAG | 0.81 | 0.76 |
| Bespoke-MiniCheck-7B | 0.79 | 0.79 |
| GPT-4.1 Mini | 0.67 | 0.61 |
| AlignScore | 0.57 | 0.68 |
| INFUSE | 0.64 | 0.58 |

### E.4 FULL HYPERPARAMETER CONFIGURATIONS

Figure 2, Figure 3, Figure 4 and Figure 5 report hard prediction results for all hyperparameter configurations tested for AlignScore, INFUSE, Llama-3.1-Bespoke-MiniCheck-7B, and RAG, respectively. Hyperparameter definitions are provided in §4.2.

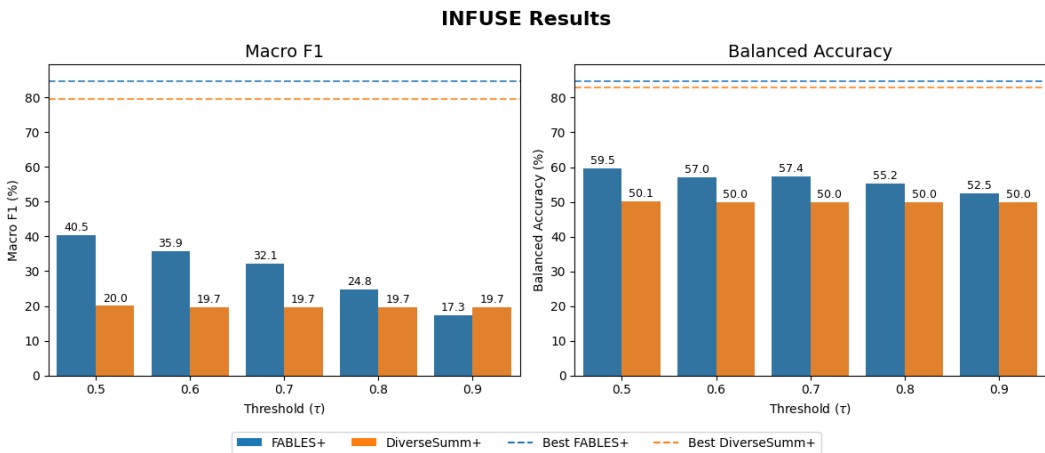

**Figure 2:** Hard prediction results for all AlignScore configurations on the FABLES+ and DiverseSumm+ datasets. We varied the threshold used to convert entailment probabilities into binary labels ($\tau$) and the number of chunk-level probabilities averaged ($k$). Each value shows the performance for a specific ($\tau$, $k$) pair.

**INFUSE Results**

**Figure 3:** Hard prediction results for all INFUSE configurations on the FABLES+ and DiverseSumm+ datasets. We varied the threshold $\tau$ used to convert entailment probabilities into binary labels. Dashed lines indicate the best result across all methods from Table 1.

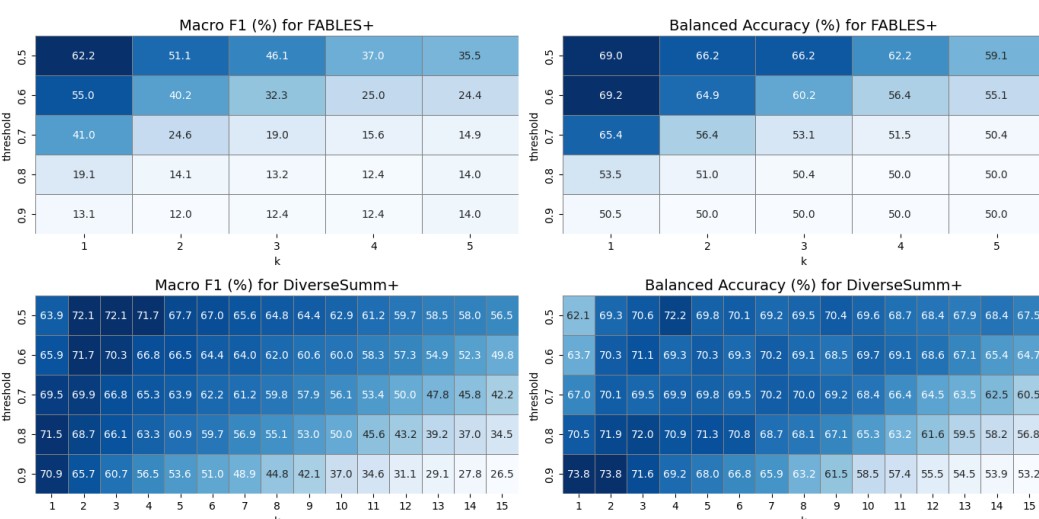

**Figure 4:** Hard prediction results for all Llama-3.1-Bespoke-MiniCheck-7B configurations on the FABLES+ and DiverseSumm+ datasets. We varied the threshold $\tau$ used to convert entailment probabilities into binary labels and the number of chunk-level probabilities averaged ($k$). Each value shows the performance for a specific ($\tau$, $k$) pair.

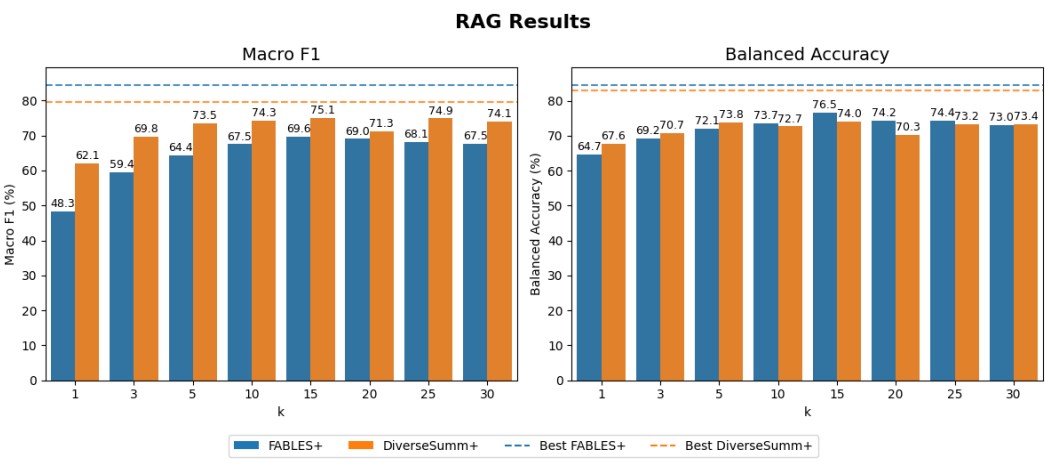

**Figure 5:** Hard prediction results for all RAG configurations on the FABLES+ and DiverseSumm+ datasets. We varied the top $k$ chunks retrieved. Dashed lines indicate the best result across all methods from Table 1.

## E.5 ALTERNATIVE VERDICT GENERATION PROMPT

Table 6 reports hard prediction results for RAG, Gemini 1.5 Pro, GPT-4.1 Mini, and VeriTrail, using two different prompts for verdict generation: (1) VeriTrail's default prompt (see Appendix C.2.2), and (2) a prompt from the original FABLES paper, shown below:

> *You are provided with a context and a statement. Your task is to carefully read the context and then determine whether the statement is true or false. Use the information given in the context to make your decision.*
>
> *Context:* {context}
>
> *Statement:* {claim}
>
> *Question: Based on the context provided, is the above statement True or False?*
>
> *Answer:*

**Table 6:** Hard prediction results (%) for the FABLES+ and DiverseSumm+ datasets using VeriTrail's prompt for Verdict Generation ("Orig.") and a prompt from the original FABLES paper ("Alt."). We report macro $F_1$, balanced accuracy (Bal. Acc.), and class-specific precision and recall for fully supported (FS) and not fully supported (NFS) claims. Bolded values indicate the better score in each Orig./Alt. pair.

| Dataset | Method | Macro $F_1$ | | Bal. Acc. | | Precision$_{FS}$ | | Recall$_{FS}$ | | Precision$_{NFS}$ | | Recall$_{NFS}$ | |
|---|---|---|---|---|---|---|---|---|---|---|---|---|---|
| | | Orig. | Alt. | Orig. | Alt. | Orig. | Alt. | Orig. | Alt. | Orig. | Alt. | Orig. | Alt. |
| FABLES+ | VeriTrail ($q=1$) | **67.9** | 66.6 | **79.0** | 70.4 | **96.1** | 92.1 | 76.7 | **84.5** | 36.6 | **37.5** | **81.2** | 56.2 |
| | GPT-4.1 Mini | 63.2 | **66.1** | 60.2 | **62.8** | 88.4 | **89.1** | **98.4** | 97.4 | **70.0** | 64.3 | 21.9 | **28.1** |
| | RAG ($k=3$) | **56.2** | 54.9 | **63.5** | 57.7 | **90.7** | 88.1 | 69.0 | **76.6** | **24.0** | 21.8 | **58.1** | 38.7 |
| | RAG ($k=5$) | **58.9** | 58.2 | **66.5** | 61.7 | **91.7** | 89.4 | 71.7 | **78.3** | **26.8** | 25.9 | **61.3** | 45.2 |
| | RAG ($k=10$) | **60.4** | 59.7 | **64.9** | 62.8 | **90.6** | 89.7 | 78.3 | **80.4** | **28.6** | 28.0 | **51.6** | 45.2 |
| DiverseSumm+ | VeriTrail ($q=1$) | **81.0** | 70.4 | **85.6** | 68.2 | **95.8** | 82.4 | 81.4 | **93.6** | 62.9 | **70.0** | **89.8** | 42.9 |
| | GPT-4.1 Mini | **60.6** | 51.8 | **59.7** | 54.4 | **78.4** | 76.0 | 92.9 | **98.6** | 56.5 | **71.4** | **26.5** | 10.2 |
| | RAG ($k=3$) | **69.0** | 54.1 | **68.8** | 54.3 | **84.8** | 77.5 | 85.4 | **90.5** | **53.5** | 38.1 | **52.3** | 18.2 |
| | RAG ($k=5$) | **71.0** | 59.1 | **70.7** | 58.4 | **85.6** | 79.1 | 86.9 | **94.2** | **57.1** | 55.6 | **54.5** | 22.7 |
| | RAG ($k=10$) | **66.7** | 58.9 | **65.3** | 58.4 | **82.6** | 79.0 | 89.8 | **96.4** | 56.2 | **64.3** | **40.9** | 20.5 |

## E.6 VERITRAIL TERMINATION CONTROL

Table 7 reports hard prediction results using different values of $q$ for VeriTrail on FABLES+ and DiverseSumm+. At most 3 iterations are possible for FABLES+ and 5 for DiverseSumm+. Therefore, their maximal $q$ values are 3 and 5, respectively; in these settings, VeriTrail always verifies at least one root node. We also include results for RAG, the best-performing baseline method, for direct comparison. All VeriTrail variants outperformed the baseline.

## E.7 ADDITIONAL MODELS

Table 8 reports hard prediction results for VeriTrail and RAG (the top-performing baseline method) with the `DeepSeek-V3`, `gemini-2.5-flash-preview-04-17`, and `mistral-large-2411` models.

**Table 7:** Hard prediction results (%) for the FABLES+ (F) and DiverseSumm+ (D) datasets for VeriTrail at varying $q$ values and RAG. For RAG, we report results using the best-performing $k$ value by macro $F_1$ ($k$=15 for F, $k$=30 for D). We report macro $F_1$, balanced accuracy (Bal. Acc.), and class-specific precision and recall for fully supported (FS) and not fully supported (NFS) claims. A dash (-) indicates that the configuration was not evaluated. Bolded values indicate the highest score in each column.

| Method | Setting | Macro F$_1$ | | Bal. Acc. | | Precision$_{FS}$ | | Recall$_{FS}$ | | Precision$_{NFS}$ | | Recall$_{NFS}$ | |
|---|---|---|---|---|---|---|---|---|---|---|---|---|---|
| | | F | D | F | D | F | D | F | D | F | D | F | D |
| VeriTrail | $q = 1$ | 69.1 | **80.7** | 80.5 | **85.5** | **96.6** | **95.7** | 77.0 | 82.4 | 38.2 | 61.9 | **83.9** | **88.6** |
| | $q = 2$ | 80.4 | 76.5 | 85.4 | 74.7 | 96.5 | 86.9 | 90.2 | 92.6 | 58.1 | 71.4 | 80.6 | 56.8 |
| | $q = 3$ | **85.7** | 76.9 | **87.6** | 73.5 | **96.6** | 85.7 | **94.5** | **97.1** | **71.4** | **84.6** | 80.6 | 50.0 |
| | $q = 4$ | - | 74.6 | - | 71.3 | - | 84.6 | - | **97.1** | - | 83.3 | - | 45.5 |
| | $q = 5$ | - | 74.6 | - | 71.3 | - | 84.6 | - | **97.1** | - | 83.3 | - | 45.5 |
| RAG | best-$k$ | 66.5 | 74.3 | 72.7 | 72.9 | 93.1 | 86.1 | 80.9 | 91.2 | 36.4 | 66.7 | 64.5 | 54.5 |

**Table 8:** Hard prediction results (%) on the FABLES+ (F) and DiverseSumm+ (D) datasets for Veri-Trail and RAG with the `DeepSeek-V3` (DeepSeek), `gemini-2.5-flash-preview-04-17` (Gemini 2.5 Flash), and `mistral-large-2411` (Mistral) models. For RAG, we use the best-performing $k$ value based on macro $F_1$: DeepSeek = 3/30, Gemini = 5/10, and Mistral = 5/10, for F/D, respectively. We report macro $F_1$, balanced accuracy (Bal. Acc.), and class-specific precision and recall for fully supported (FS) and not fully supported (NFS) claims. Bolded values indicate the best-performing method for each dataset and metric.

| Model | Method | Macro F$_1$ | | Bal. Acc. | | Precision$_{FS}$ | | Recall$_{FS}$ | | Precision$_{NFS}$ | | Recall$_{NFS}$ | |
|---|---|---|---|---|---|---|---|---|---|---|---|---|---|
| | | F | D | F | D | F | D | F | D | F | D | F | D |
| DeepSeek | VeriTrail ($q = 1$) | **61.7** | **68.0** | **70.9** | **73.2** | **93.4** | **89.8** | 73.1 | 68.8 | **29.7** | 46.3 | **68.8** | **77.6** |
| | RAG | 59.4 | 66.0 | 63.1 | 63.9 | 90.0 | 80.1 | **79.3** | **97.2** | 27.3 | **78.9** | 46.9 | 30.6 |
| Gemini 2.5 Flash | VeriTrail ($q = 1$) | **70.0** | **69.9** | **80.2** | 72.3 | **96.2** | **87.2** | 79.2 | 77.3 | **39.4** | 50.8 | **81.2** | **67.3** |
| | RAG | 66.6 | 68.9 | 73.2 | 67.5 | 93.4 | 82.5 | **80.7** | **90.1** | 36.2 | **61.1** | 65.6 | 44.9 |
| Mistral | VeriTrail ($q = 1$) | 49.2 | 67.0 | **67.2** | **73.8** | **95.5** | **91.7** | 49.7 | 64.7 | 20.6 | 44.8 | **84.6** | **83.0** |
| | RAG | **55.2** | **67.6** | 62.1 | 66.5 | 90.8 | 82.2 | **70.4** | **88.2** | **21.9** | **56.8** | 53.8 | 44.7 |

## F    NLI BASELINE DETAILS

In §4.2, we provided an overview of the NLI baselines evaluated in our experiments: AlignScore (Zha et al., 2023), INFUSE (Zhang et al., 2024a), and Llama-3.1-Bespoke-MiniCheck-7B (Bespoke Labs, 2024). We implemented all methods using the official repositories and default hyperparameter settings. For AlignScore and INFUSE, the only modification we made was applying our sentence-splitting method (described in §3.1.2) to ensure consistency with VeriTrail.

For AlignScore, we used the top-performing model from the original paper: AlignScore-large (355M parameters), which is based on RoBERTa-large (Liu et al., 2019). For INFUSE, we used the same NLI model[11] as the original paper: an ALBERT-xlarge model (Lan et al., 2020), fine-tuned on MNLI (Williams et al., 2018) and VitaminC (Schuster et al., 2021), with 58.7M parameters. We accessed Llama-3.1-Bespoke-MiniCheck-7B via Hugging Face[12]. The model is based on InternLM2.5-7B-Chat (Cai et al., 2024), fine-tuned on ANLI (Nie et al., 2020) and synthetically-generated data.

For FABLES+, each claim was compared to a single document: the corresponding book. For DiverseSumm+, which contains multiple documents (i.e., articles), we tested all pairwise combinations of documents and claims. For AlignScore and Llama-3.1-Bespoke-MiniCheck-7B, we selected the top $k$ chunks (across a single document for FABLES+; across all documents for DiverseSumm+) with the highest probability of entailment for a given claim. For INFUSE with DiverseSumm+, we used the document with the highest entailment probability for each claim.

AlignScore and INFUSE were run for 12 and 24 hours, respectively, on an NVIDIA A10G GPU. Llama-3.1-Bespoke-MiniCheck-7B was run for 20 hours on an NVIDIA A100 GPU.

## G    LIMITATIONS

To better understand VeriTrail's limitations, we manually reviewed 60 error cases per dataset. For each dataset, we randomly sampled claims from the $q = 1$ and $q = 3$ settings, evenly split between false negatives (claims incorrectly labeled as "Fully Supported") and false positives (claims incorrectly labeled as "Not Fully Supported"). All analyzed outputs were produced using gpt-4o-2024-08-06.

We identified two primary causes of false negatives. The first was invalid inferences. For example:

- **Claim (from FABLES+):** Charlie Brown is eager to reunite with her husband Harry
- **Assigned Label:** Fully Supported
- **Model's Reasoning:** "*The evidence strongly supports the idea that Charlie is eager to reunite with Harry... The evidence does not explicitly state that Harry is her husband, but it is implied through the anticipation of reunion.*"
- **Our Analysis:** Charlie's excitement about reuniting with Harry does not, on its own, imply that he is her husband.

The second cause of false negatives was overlooking components of the claim. For instance:

- **Claim (from DiverseSumm+):** NATO and its member states have supplied weapons to bolster Ukraine's defense capabilities [in response to Russia's aggression]
- **Assigned Label:** Fully Supported
- **Model's Reasoning:** "*The evidence from the excerpts clearly indicates that NATO member states, such as the United States and various EU countries, have supplied weapons and military aid to Ukraine... The evidence does not explicitly mention NATO as an organization supplying weapons, but it does strongly imply that NATO member states are actively involved...*"
- **Our Analysis:** The evidence supports the role of member states, but not NATO as an organization. The model acknowledges this fact but ultimately glosses over it in its final judgment.

---

[11]https://huggingface.co/tals/albert-xlarge-vitaminc-mnli
[12]https://huggingface.co/bespokelabs/Bespoke-MiniCheck-7B

For false positives, the sources of error were more varied. One example was omission of relevant context from the evidence:

- **Claim (from DiverseSumm+):** Nike's apparel units in North America declined by high-single digits

- **Assigned Label:** Not Fully Supported

- **Model's Reasoning:** In the Evidence Selection step, the model correctly selected the following text as evidence: "*Region where Nike saw a decline in apparel units by high-single digits.*" However, it failed to include a related header, "*NORTH AMERICA (GEO):*", that appeared several lines earlier. As a result, it generated the following evidence summary: "*Sentence 12 states that Nike saw a decline in apparel units by high-single digits in a region.*" In the Verdict Generation step, it reasoned as follows: "*The only relevant sentence indicates a decline in apparel units by high-single digits in a region but does not specify that the region is North America.*"

- **Our Analysis:** The model identified the correct evidence, but missed the broader context needed to interpret it. Our Evidence Selection prompt instructs the model to include sentences that provide critical context; however, this example demonstrates that identifying such sentences can be challenging when they are dispersed or presented in an ambiguous form (e.g., sentence fragements).

We also observed differences in error types across the datasets. In FABLES+, some error cases involved multi-hop reasoning (i.e., combining multiple pieces of evidence, often located in different parts of the source text, to support or refute a claim). For example:

- **Claim (from FABLES+):** "Weyward" chronicles the interconnected narratives of three women named Altha, Violet, and Kate

- **Assigned Label:** Not Fully Supported

- **Model's Reasoning:** "*The excerpts clearly mention narratives involving Altha, Violet, and Kate... However, the excerpts do not provide strong evidence that these narratives are interconnected.*"

- **Our Analysis:** This claim is difficult to verify because the evidence linking the three characters is subtle and dispersed throughout the story. It is unlikely that a single Evidence Selection step would surface all connections needed to fully support the claim.

In DiverseSumm+, we did not observe multi-hop reasoning errors. This is likely because its source material consists of self-contained articles on distinct topics, as opposed to a long narrative with interdependent parts.

A unique error type in DiverseSumm+ was the use of parametric knowledge, presumably due to the real-world nature of the source material. For instance:

- **Claim (from DiverseSumm+):** NVIDIA collaborates with other tech giants [besides itself]

- **Assigned Label:** Fully Supported

- **Model's Reasoning:** "*The evidence ... clearly indicates that NVIDIA has a partnership with Alphabet Inc.'s Google Cloud Platform, which is a tech giant. Sentence 45 suggests that Oracle provides cloud infrastructure for NVIDIA's DGX AI supercomputer, and Oracle is generally considered a tech giant.*"

- **Our Analysis:** The model correctly identified relevant collaborations. However, the designation of Alphabet and Oracle as technology giants was neither explicitly stated nor implied by the evidence and likely reflects reliance on parametric knowledge.

Collectively, these findings highlight several opportunities for future work, including mitigating common reasoning errors (e.g., invalid inferences, reliance on parametric knowledge, etc.), supporting multi-hop claim verification, and exploring how the nature of the source material affects error patterns. We also flag that VeriTrail is not tied to using an LM for verification. The Verdict Generation step is modular and can be replaced with any verifier that assesses a claim's faithfulness based on evidence, so experimentation with alternative verifiers is another opportunity for future work.

## H   ERROR STAGE ANALYSIS

In this section, we address two questions:

1. Which stages are the most common sources of hallucination in the MGS processes we evaluated: hierarchical summarization (used in FABLES+) and GraphRAG (used in DiverseSumm+)?

2. How consistent is error stage identification across VeriTrail variants? For instance, if we run VeriTrail with $q = 1$ instead of $q = 3$ on a set of claims, how similar are the resulting error stage distributions?

To answer these questions, we analyzed the VeriTrail variants used in our ablation studies, covering different input size limits (Appendix E.2), confidence thresholds (Appendix E.3), values of $q$ (Appendix E.6), and models (Appendix E.7). We included 13 variants for FABLES+ and 15 for DiverseSumm+.

For each variant, we identified its **true positive claims**, defined as the set of claims that met all of the following conditions:

- The claim was from the subset of FABLES+ or DiverseSumm+ described in Appendix E (since not all variants were evaluated on the full datasets);

- The claim was correctly labeled "Not Fully Supported"; and

- At least one error stage was identified for the claim (see §3.2 for cases where error stage identification is not possible).

The average number of true positive claims per variant was 24 for FABLES+ and 32 for DiverseSumm+.

### H.1   WHICH STAGES ARE MOST PRONE TO HALLUCINATION?

For a given VeriTrail variant $v$ and a possible error stage $s$, let:

$$p_{v,s} = \frac{\text{\# true positive claims for } v \text{ where } s \text{ was identified as an error stage}}{\text{\# true positive claims for } v}.$$

In other words, stages with a higher value of $p$ were more frequently identified as sources of hallucination by variant $v$.

We computed $p_{v,s}$ for all combinations of $v$ and $s$. As noted in Appendix B.1.1 and Appendix B.2.1, there are 4 stages in FABLES+ and 6 in DiverseSumm+. However, root nodes (stage 1) cannot be the source of hallucinations, leaving 3 and 5 possible error stages, respectively.

To aggregate across variants, we computed the weighted average for each stage $s$:

$$\bar{p}_s = \frac{\sum_v n_v \cdot p_{v,s}}{\sum_v n_v}$$

where $n_v$ is the number of true positive claims for variant $v$. This weighting reflects the intuition that variants with more true positives are likely more reliable and should have greater influence on the overall estimate.[13]

Let $C_{TP}$ denote the set of all true positive claims across all variants. To estimate uncertainty, we applied bootstrap resampling over $C_{TP}$. Specifically, we performed 1,000 iterations in which we sampled with replacement the same number of claims as in $C_{TP}$ and recomputed $\bar{p}_s$ for each stage. The resulting distribution was used to compute 95% confidence intervals. Final estimates for both datasets are reported in Table 9.

---

[13]We also tested unweighted averages and observed negligible differences.

**Table 9:** Error attribution rate ($\bar{p}_s$) for each possible error stage in FABLES+ and DiverseSumm+. This metric captures the proportion of correctly identified hallucinated claims (i.e., true positives) for which a given stage was identified as a likely source of error. Values are averaged across VeriTrail variants, weighted by the number of true positives per variant. 95% confidence intervals were estimated via bootstrap resampling. Bolded rows indicate the most frequently implicated stage for each dataset.

| Dataset | Stage | Error Attribution Rate (95% CI) |
|---|---|---|
| FABLES+ (Hierarchical Summarization) | 2 | 0.20 [0.16, 0.24] |
| | 3 | **0.47 [0.42, 0.53]** |
| | 4 | 0.32 [0.27, 0.37] |
| DiverseSumm+ (GraphRAG) | 2 | 0.15 [0.12, 0.18] |
| | 3 | 0.09 [0.07, 0.12] |
| | 4 | **0.41 [0.36, 0.45]** |
| | 5 | 0.13 [0.10, 0.16] |
| | 6 | 0.22 [0.19, 0.26] |

## H.2 HOW CONSISTENT IS ERROR STAGE IDENTIFICATION?

We assessed the consistency of error stage distributions across VeriTrail variants using two metrics:

1. **Jensen-Shannon Divergence** (JSD; Lin, 1991) measures the similarity between two probability distributions. It is bounded between 0 and 1, with lower values indicating greater similarity. We computed the JSD for all pairs of variants and observed a mean JSD of 0.02 for FABLES+ and 0.03 for DiverseSumm+, suggesting that the error stage distributions were highly consistent across variants.

2. **Spearman Rank Correlation** (Spearman, 1961) measures the similarity between rankings assigned to a set of items – in our case, the relative frequency of different error stages. It ranges from -1 (perfect inverse agreement) to 1 (perfect agreement). Across all pairwise comparisons, the mean correlation was 0.67 for FABLES+ and 0.66 for DiverseSumm+, indicating substantial agreement in the ranking of stages across variants.

