# OpenReview forum: "VeriTrail: Closed-Domain Hallucination Detection with Traceability"
_ICLR.cc/2026/Conference — ICLR 2026 Poster_

### Official Review · Reviewer_xtw1 · 2025-10-31

**Soundness:** 2
**Presentation:** 3
**Contribution:** 2
**Rating:** 2
**Confidence:** 3

**Summary:**

**Summary**

The paper introduces an algorithm for detecting hallucinations and provide traceability in processes with multiple generative steps (MGS). Processes with multiple generative steps involve multiple calls to an LLM and typically involve feeding output of a step as the input of a subsequent step. In the context of the paper, traceability includes identifying the source from which a claim was derived and also, in which subprocess any error(s) were introduced.

**Contributions**
- An algorithm for detecting hallucinations in processes with MGS
- Two new datasets for faithfulness of LLM outputs that include outputs of intermediate steps

**Strengths:**

- The paper addresses an important and underserved area of combating hallucinations in LLM outputs. Namely, in complex tasks that involve several LLM calls with potential chaining from output of one call to the input of another.
- I commend the clarity with which they have described their framework.
- I understand that in newer research directions, it can be challenging to find strong benchmarks/baselines to compare. So, I appreciate the effort in making their own datasets (FABLES+, DiverseSumm+).

**Weaknesses:**

- As the authors state at the end of the paper, VeriTrail is a reference-based method, meaning that it relies on relevant source text and compares claims with this source text. So, its applicability is inherently limited to transformative tasks like summarization (and not, say, to question-answering when the LLM is relying on its internal knowledge). Even the benchmarks that are used in the experiments are both essentially summarization tasks. While I don't mind a method limited in applicability, this limitation should be presented clearly.
- None of the baselines assume or use the DAG of the generative process with MGS. IMO, this makes them weak baselines for comparison.
- The prompts used in VeriTrail have some examples or clarifications. How were these chosen? The effect of these on the performance of VeriTrail is unclear. I am afraid that if they were added in an ad-hoc fashion, would these necessarily translate to other tasks or benchmarks?

Ultimately, I am giving the paper a score of 2 due to the above reasons.

**Questions:**

- With regard to the weakness 1 mentioned above, do you think VeriTrail is applicable to tasks other than summarization? If yes, please mention them and how would you apply VeriTrail in such tasks?
- What was the rationale for each step in VeriTrail being done with a large monolithic prompt? Have you considered breaking it down into smaller steps with more specific instructions? I am asking this because I see that the prompts in the appendix are written in a step-by-step fashion, asking the LM to do one after the other in a single prompt.
- I am curious to see if using something like DSPy [1] for optimizing the prompts would lead to better performance.

**References:**

[1] DSPy: Compiling Declarative Language Model Calls into Self-Improving Pipelines (Khattab et al: [https://arxiv.org/abs/2310.03714](https://arxiv.org/abs/2310.03714))

---

> ### Author Response · Authors · 2025-11-21
>
> # Response Part 1
> We thank the reviewer for their comments and appreciate the time they spent reviewing our work. We address each point below.
>
> **(1) Is VeriTrail “applicable to tasks other than summarization?” If yes, “please mention them” and describe “how would you apply VeriTrail in such tasks." If not, “this limitation should be presented clearly.”**
>
> - VeriTrail is not limited to summarization. It applies to any closed-domain task (i.e., a language model is given source material and asked to produce output faithful to it). This includes, but is not limited to, summarization, question answering based on provided text, content creation (e.g., drafting documentation from notes), information extraction (e.g., identifying entities and relationships in text), and evidence-based analysis (e.g., assessing whether requirements are met given supporting documents). All of these tasks use text as input and produce text as output, so the generative processes used to execute the tasks can be represented as DAGs, claims can be extracted from the final outputs, and the extracted claims can be verified by VeriTrail.
>
> - We do not view VeriTrail’s focus on closed-domain tasks as a limitation. Firstly, prior works routinely distinguish between closed-domain and open-domain hallucination detection, and many methods target only one of these settings. Secondly, closed-domain tasks are among the most common and increasingly important applications of language models, so improving hallucination detection and enabling traceability specifically for such tasks is clearly useful.
>
> - VeriTrail’s scope (closed-domain hallucination detection) is already defined explicitly in the title and the introduction. Nonetheless, in the updated version of the paper, we have emphasized in Section 5 that VeriTrail is designed for closed-domain hallucination detection and is not intended for open-domain hallucination detection.
>
> **(2) The benchmarks used in the experiments “are both essentially summarization tasks.” Will VeriTrail generalize to other tasks?**
>
> - The claim that we only tested on summarization tasks is inaccurate and does not reflect the full scope of our evaluation. Firstly, we tested VeriTrail on two distinct tasks in NLP: summarization (FABLES+) and question answering (DiverseSumm+). These tasks differ in input-output format, objectives (compressing content vs. answering a specific query), and failure modes (e.g., QA may require determining that a question cannot be answered from the source material). Secondly, we also evaluated VeriTrail across different types of MGS processes (hierarchical summarization and GraphRAG) and source texts (fiction books and news articles), providing additional evidence of generalizability.
>
> - Constructing FABLES+ and DiverseSumm+ required extensive manual effort and cost. Creating additional benchmarks of comparable size and complexity is a significant undertaking beyond the scope of a single paper.
>
> **(3) “None of the baselines use the DAG of the generative process,” making them “weak baselines for comparison.”**
>
> - This is incorrect. Section 4.4 refers to Appendix E.1 which describes an ablation study where we created and evaluated a variant of RAG (the strongest baseline method) that uses the DAG of the generative process. VeriTrail outperformed this variant (see VT-RAG in Table 3).
>
> - More broadly, we disagree that baselines must use the DAG in order to be sufficiently strong for comparison. Firstly, we believe that if a closed-domain hallucination detection method is widely used and known to perform well, it is a good baseline for comparison. Our baselines satisfy this requirement, as evidenced by their competitive performance in our experiments and in prior benchmarks (e.g., see Section 4.2 for discussion of the NLI baselines). Secondly, to the best of our knowledge, there are no prior closed-domain hallucination detection methods that use a DAG as input or attempt to trace through the intermediate outputs of an MGS process in some other way. VeriTrail’s tracing mechanism is precisely what makes it unique and innovative. Our evaluation is therefore designed to assess the impact of tracing by comparing against baselines that lack this capability.

---

> ### Author Response · Authors · 2025-11-21
>
> # Response Part 2
>
> **(4) If the examples and clarifications in VeriTrail’s prompts were “added in an ad-hoc fashion,”  VeriTrail might not generalize to other benchmarks.**
>
> - Our prompts are benchmark-agnostic. They define general concepts relevant to hallucination detection (e.g., what constitutes a “Fully Supported” claim) and do not rely on benchmark-specific examples or instructions. Most importantly, our experiments indicate that VeriTrail generalizes well: we used the same set of prompts across different tasks, MGS processes, and source texts, and VeriTrail consistently outperformed the baselines.
>
> **(5) Why not break prompts into multiple smaller prompts? Can the prompts be optimized further?**
>
> - Our prompts follow well-established techniques, such as few-shot learning ([1] Brown et al., 2020), chain-of-thought prompting ([2] Wei et al., 2022), and decomposition prompting ([3] Khot et al., 2022). We avoid further breaking up the prompts for two reasons: (1) additional prompts increase cost and latency by requiring extra model calls, and (2) splitting steps makes it difficult to preserve context (i.e., what the model produced previously and how its output will be used next), which can weaken the model’s ability to maintain coherent reasoning and make optimal decisions at each step.
>
> - The prompts are not central to our contribution. For example, as noted in Section 4.3, we intentionally used the same verdict prompt for VeriTrail and all LM-based baselines to avoid differences in performance due to prompt engineering. This allowed us to isolate the impact of VeriTrail’s key innovations: the DAG representation and traversal algorithm that enable traceability.
>
> - Even without any prompt optimization, VeriTrail already outperforms strong and diverse baseline methods. We welcome future work that explores optimization, but we believe such modifications would only reinforce, not change, our findings.
>
>
> We thank the reviewer again for their comments. We respectfully note that there seems to be some misunderstanding of the method or experiments. We hope that the explanations above are helpful and that the updated understanding will be reflected in the final assessment.
>
> ---
>
> References:
>
> [1] Brown, Tom, et al. "Language models are few-shot learners." Advances in neural information processing systems 33 (2020): 1877-1901.
>
> [2] Wei, Jason, et al. "Chain-of-thought prompting elicits reasoning in large language models." Advances in neural information processing systems 35 (2022): 24824-24837.
>
> [3] Khot, Tushar, et al. "Decomposed prompting: A modular approach for solving complex tasks." arXiv preprint arXiv:2210.02406 (2022).

---

### Official Review · Reviewer_8wrP · 2025-11-01

**Soundness:** 3
**Presentation:** 3
**Contribution:** 3
**Rating:** 6
**Confidence:** 4

**Summary:**

In this paper, the authors analyze hallucination detection in generations of autoregressive LLMs in a closed-domain setting. As opposed to just single-step generation (such as with RAG), the paper analyses hallucinations in a more complex and challenging setting, involving multiple generative steps (MGS). Toward this, the authors propose VeriTrail, which assesses the faithfulness, as well as provenance tracing and error localization using a multi-stage graph-based framework. The method begins with the final output and uses backward traversal to identify the stage at which hallucinations may have been introduced. The paper further introduces two new datasets FABLES+ and DiverseSumm+, and demonstrates the efficacy of VeriTrail over existing baselines like RAG and long-context models toward hallucination detection.

**Strengths:**

1) The paper studies a pertinent problem of detecting hallucinations in the generations of autoregressive LLMs, while also focusing on provenance tracing and error localization with multiple-generation steps, which is highly relevant given complex, multi-agent systems that have gained prominence in recent times. By identifying the intermediate stage at which hallucinations are introduced in a closed-domain setting, specific and actionable modes of improvement can be identified and undertaken in these complex systems in a transparent manner.

2) The DAG-based framework adopted by VeriTrail is principled, and helps assess the veracity of claims and sub-claims in a systematic manner. Furthermore, selective verification helps reduce unnecessary and wasteful computation over nodes that correspond to uninformative texts relative to the primary claim.

3) A significant and notable contribution of the paper is the introduction of two new datasets, FABLES+ and DiverseSumm+, that are specific to the MGS setting, and include intermediate outputs as well as human-annotated evaluations on faithfulness. Furthermore, the proposed method VeriTrail is shown to be effective, and out-performs strong existing baselines on these datasets.

**Weaknesses:**

1) The proposed method VeriTrail is not simple, and requires a fairly complex setting up of the generative process as a DAG, and ensure that each intermediate output is carefully produced and analyzed in a multi-step manner with verification by an LLM. This also places a constraint on its scalability in practice for real-time applications, given the need for multiple LLM calls for each claim/sub-claim in the final output. Could the authors also kindly provide some metrics with respect to the actual average runtime required for the overall verification pipeline, and compare with more standard approaches like RAG itself?


2) This complexity also manifests in the computational overheads and costs involved for improved performance. For instance, as presented in Appendix D, when comparing with the same LLM model (gpt-4o-2024-0806), VeriTrail  appears to cost 5x-9x more than RAG itself. These added costs can further limit practical usage in several real-world settings.


3) The quality of VeriTrail's analysis depends critically on the decomposition of the final output into factual claims, using Claimify. Given this critical dependance, if Claimify makes an ambiguous phrasing of a claim, or introduces an error in itself, the proposed method cannot recover in a meaningful manner.


4) Stage-assignment analysis: Must this setup for stage-assignment logic be explicitly programmed by a developer for each new MGS system, or does the framework offer any mechanism to automatically infer or propose a logical stage hierarchy from the process graph itself? Furthermore, how sensitive is the accuracy and utility of the 'error stage' localization to the granularity of this breakdown? Would this be more dependent on the actual facts presented, as opposed to the stage-specific assignments given the closed-domain setup?

**Questions:**

Kindly refer to the questions mentioned in the weaknesses section above. I would be happy to raise my score further if these could be adequately addressed.

---

> ### Author Response · Authors · 2025-11-21
>
> # Response Part 1
>
> We thank the reviewer for their comments and appreciate the time they spent reviewing our work. We address each point below.
>
> **(1) What is the average runtime of VeriTrail compared to RAG?**
>
> - The table below shows average per-claim runtimes in seconds for RAG and VeriTrail, using GPT-4o for verdict generation. Note that RAG consists of two parts: (1) a one-time embedding of all source text chunks, and (2) per-claim processing (embedding the claim, retrieving k chunks, and generating a verdict). We compute the amortized per-claim runtime, which divides the one-time embedding runtime across all claims and adds the result to the average per-claim processing time. For completeness, the one-time embedding runtime for RAG (averaged across runs) was 215.8 seconds for DiverseSumm+ and 45.6 seconds for FABLES+.
>
>     | Method     | Hyperparameter | FABLES+ | DiverseSumm+ |
>     |------------|----------------|---------|----------------|
>     | RAG        | k = 5          | 26.3    | 19.4           |
>     |            | k = 15         | 30.1    | 18.9           |
>     |            | k = 25         | 45.0    | 20.9           |
>     | VeriTrail  | q = 1          | 180.4   | 181.4          |
>     |            | q = 2          | 206.7   | 210.0          |
>     |            | q = 3          | 212.5   | 262.6          |
>
> - RAG is faster than VeriTrail. However:
>    - The comparison is not apples-to-apples. RAG produces only a hallucinated/not-hallucinated verdict, whereas VeriTrail also constructs an evidence trail through intermediate outputs to provide traceability. For instance, as noted in Appendix D, RAG evaluates only ~3K root nodes in DiverseSumm+, whereas VeriTrail evaluates an additional 110K intermediate nodes. Given VeriTrail’s richer functionality and significantly larger verification burden, higher runtime is expected.
>    - It is important to recognize that there are many practical applications where traceability is critical and runtime is not. For example, during evaluation, quality control, and debugging of generative processes, it is necessary to understand where hallucination was likely introduced in order to address it. Another example is high-stakes domains, such as medicine and law, where practitioners must be able to inspect the provenance of model-generated statements in order to trust them. RAG cannot satisfy these use cases. VeriTrail is the first method to provide error localization and provenance for MGS processes.
>
> **(2) VeriTrail is more expensive than RAG when using the same model (e.g., GPT-4o).**
>
> - This is expected: VeriTrail performs significantly more verification work in order to provide traceability. However, as shown in Appendix D, VeriTrail’s costs can be reduced by using a cheaper model and/or a lower value of the q hyperparameter. For example, with Gemini-2.5-Flash and q=1, VeriTrail costs only \\$0.09 - \\$0.14 per claim, while still outperforming the baselines.
>
> **(3) Errors or ambiguities in the claims extracted by Claimify may compromise VeriTrail’s verification results.**
>
> - VeriTrail is not tied to Claimify specifically: it can work with any claim extraction system. We used Claimify in our experiments due to its strong performance.
>
> - VeriTrail handles ambiguous claims as follows: the verdict prompt (Appendix C.2.2) asks the model to identify whether a claim has multiple possible interpretations and, if so, to list them and select the interpretation that most readers would likely agree with. This approach allows VeriTrail to proceed with verification while making the ambiguity and the chosen interpretation transparent to the user.
>
> - Handling inaccurate claims (i.e., claims that misrepresent the text from which they were extracted) is outside of VeriTrail's scope. We follow prior works, which consistently treat claim extraction as independent from claim verification and assume that the claims provided to the verification method are accurate (e.g., [1] Min et al., 2023; [2] Hu et al., 2025). Therefore, we view the evaluation of claim extraction quality as an upstream task, not part of our verification method.
>
> ---
> References:
>
> [1] Min, Sewon, et al. "Factscore: Fine-grained atomic evaluation of factual precision in long form text generation." Proceedings of the 2023 Conference on Empirical Methods in Natural Language Processing. 2023.
>
> [2] Hu, Qisheng, Quanyu Long, and Wenya Wang. "Decomposition Dilemmas: Does Claim Decomposition Boost or Burden Fact-Checking Performance?." Proceedings of the 2025 Conference of the Nations of the Americas Chapter of the Association for Computational Linguistics: Human Language Technologies (Volume 1: Long Papers). 2025.

---

> > ### Author Response · Authors · 2025-11-21
> >
> > # Response Part 2
> >
> > **(4) Does the stage-assignment logic need to be explicitly programmed for each new MGS process? Is error-localization accuracy sensitive to stage assignment?**
> >
> > - No, individual programming for each new MGS process is not required. By default, VeriTrail can infer stages directly from the DAG structure as follows: nodes with no source nodes (i.e., no incoming edges) are stage 1; nodes whose source nodes include a stage 1 node are stage 2; nodes whose source nodes include a stage 2 node are stage 3; etc. We used this procedure for hierarchical summarization (FABLES+), as described in Appendix B.1.1. VeriTrail also supports custom stage definitions, which is helpful when a process has well-defined step types, such as GraphRAG (DiverseSumm+).
> >
> > - No, error-localization accuracy is not sensitive to stage assignment. VeriTrail’s verification process (i.e., evidence selection and verdict generation) operates solely on the DAG structure and node contents; stage labels are not used. After verification is complete, VeriTrail identifies the nodes that contributed evidence in the last iteration where the claim was deemed “Fully Supported,” and their stages are reported as the error stages (see Section 3.2). Changing the stage schema does not affect which nodes are implicated, only how they are grouped and described.

---

### Official Review · Reviewer_xoWU · 2025-11-03

**Soundness:** 3
**Presentation:** 3
**Contribution:** 2
**Rating:** 6
**Confidence:** 4

**Summary:**

This paper addresses closed-domain hallucination in complex, multi-step generative (MGS) processes, where error risk is amplified. The authors argue that simple detection is insufficient for MGS and propose "traceability", the ability to pinpoint where errors are introduced. They present a novel method that models the generation process as a graph to achieve both detection and traceability. To evaluate this, new datasets with intermediate steps were created. Experiments show the proposed method outperforms baselines in detection while also providing this novel traceability.

**Strengths:**

The paper tackles the critical problem of error propagation in MGS processes. The focus on "traceability" beyond simple detection is a key conceptual contribution.

The method is tested against strong, comprehensive baselines.

**Weaknesses:**

A primary weakness of this method is its recursive reliance on Large Language Models (LLMs). Core steps of VeriTrail, including sub-claim decomposition, evidence selection, and final verdict generation, are all dependent on LLMs. This creates a fundamental "verifier's dilemma": the LLM used for verification may itself hallucinate or make faulty inferences.

The paper's own error analysis in Appendix G concedes that the verification model itself can make "invalid inferences" or improperly use "parametric knowledge." This directly undermines the claimed reliability of the detection method, as the verification process itself may introduce new errors.

**Questions:**

Please refer to the previous section.

---

> ### Author Response · Authors · 2025-11-21
>
> We thank the reviewer for their comments and appreciate the time they spent reviewing our work. We address each point below.
>
> **(1) VeriTrail uses a language model (LM) to verify whether a claim is hallucinated, but the LM may itself hallucinate or make reasoning errors.**
>
> - VeriTrail is not tied to using an LM for verification. The verdict-generation step is modular and can be replaced with any verifier that assesses a claim’s faithfulness based on evidence. VeriTrail’s key components (i.e., the DAG representation and traversal algorithm that enable traceability) are agnostic to the choice of verifier. In fact, to ensure that any performance differences between VeriTrail and the baselines were not attributable to the choice of verifier, we intentionally used the same verdict-generation prompt for all methods.
>
> - We agree that LM-based verifiers can make mistakes, but it is important to recognize that there is no such thing as a "perfect" verifier: there will always be cases where a human (or another verifier) disagrees with the reasoning or the verdict (e.g., due to different ways of handling conflicting or ambiguous evidence). Therefore, we believe it is critical for a hallucination detection method to make its decisions transparent and easy to review. This is exactly what VeriTrail does: it not only achieves strong hallucination detection performance and outperforms the baselines, but it also produces an evidence trail for every verdict and, importantly, it guarantees that none of the sentences in the evidence trail are hallucinated (see Section 3.1.2).

---

### Official Review · Reviewer_1wLF · 2025-11-03

**Soundness:** 4
**Presentation:** 3
**Contribution:** 4
**Rating:** 8
**Confidence:** 3

**Summary:**

This paper proposes a hallucination detection framework for multi-step generation process. It models the tracibility of generated content (the chain of rationale) with a directed acyclic graph and convert the hallucination detection of the final problem to a traverse on graph. Experiments on book summary and news summary shows that it outperforms previous method.

**Strengths:**

+ The paper proposes a new framework for hallucination detection in multi-generation process.
+ It curates two datasets for faithfulness evaluation in MGS based on previous benchmarks.
+ The proposed VeriTrail framework shows a substaintial improvement over previous baseline.

**Weaknesses:**

+ Missing intermediate results. The ablation analysis In Appendix E shows that evidence selection using language model plays an important role in final performance so it would be necessary to include the accuracy of the evidence selection in evaluation.

+ Since VeriTrail uses stage-by-stage verfiication and does not have to always checks the source materials if the verfiication process exits early, I would suggest analyzing the API token cost of VeriTrail and comparing with the baseline method.

+ a minor comment is to use vector graph in Figure 1 for better readability.

**Questions:**

See the weaknesses above.

---

> ### Author Response · Authors · 2025-11-21
>
> We thank the reviewer for their comments and appreciate the time they spent reviewing our work. We address each point below.
>
> **(1) Accuracy of evidence selection was not evaluated.**
>
> - We agree that evaluating the accuracy of the evidence selection step could potentially be useful. However, the document collections in our benchmarks are very large, and a single claim may have dozens or even hundreds of potentially relevant pieces of evidence spread across many documents. Creating human-annotated ground truth for all relevant evidence in such a large corpus would be prohibitively costly, time-consuming, and likely incomplete. We also avoided using LMs to evaluate evidence selection, as doing so would have introduced circularity. For these reasons, we prioritized evaluating hallucination detection performance.
> - Our ablation study in Appendix E.1 indicates that evidence selection is effective and meaningfully contributes to VeriTrail’s overall performance. To isolate the effect of evidence selection, we compared VeriTrail against a variant that retained the verdict prompt and tracing mechanism but replaced evidence selection with RAG-style embedding-based retrieval. VeriTrail outperformed this variant across both datasets.
>
> **(2) “I would suggest analyzing the API token cost of VeriTrail.”**
>
> - This analysis is provided in Appendix D.

---

### Author Response · Authors · 2025-11-21

We thank all reviewers for their feedback. We appreciate that the reviewers recognized the importance of the problem addressed by our work, the novelty of our contributions, and the strength of our empirical results.

We have responded to each reviewer’s comments in detail. We have also uploaded an updated version of the paper. The main changes are: (a) adjusting the RAG cost estimates in Appendix D to reflect that the one-time corpus-embedding cost can be amortized across claims, and (b) expanding Section 5 (Related Work) to include a discussion of the difference between open-domain and closed-domain hallucination detection and emphasizing our focus on the latter.

---

### Meta-Review · Area_Chair_CBc5 · 2026-01-08

**Summary:**

Reviewers’ concerns centered on (i) practicality and cost—VeriTrail requires many LLM calls over many intermediate nodes, leading to substantially higher runtime/expense than standard RAG-style detectors, which could limit real-world scalability; (ii) the “verifier’s dilemma”—core steps (decomposition, evidence selection, verdicting) depend on an LM that can itself hallucinate or reason incorrectly, raising questions about the reliability of the verification process; (iii) component-level transparency/diagnostics—some reviewers wanted intermediate metrics such as evidence-selection accuracy and more detailed error analysis to understand what drives end-to-end gains; and (iv) scope and baseline fairness—one reviewer argued the method seemed limited to summarization and that baselines were weak because they didn’t exploit the MGS DAG structure, which would affect how compelling the claimed improvements are. These concerns, weighed against generally positive assessments of novelty and empirical gains, informed my suggested accept with the main caveat that the method is best positioned for closed-domain, auditability-critical settings rather than latency-sensitive deployment.

**Reviewer Concerns:**

Addressed by the rebuttal: (1) Scope/applicability: the authors clarified VeriTrail targets closed-domain settings and is applicable beyond summarization (they cite text-grounded QA and other document-based generation pipelines), so the “summarization-only” critique is largely resolved. (2) Baseline fairness / “no DAG baselines”: they point to an explicit DAG-aware ablation/baseline variant (VT-RAG) and argue it isolates the benefit of tracing, addressing the claim that comparisons were inherently unfair. (3) Cost/runtime questions: they added/clarified token-cost and runtime analyses (including amortization of one-time embedding costs for RAG) and explain the overhead is expected given many more nodes are verified for traceability. (4) Prompt concerns: they argue prompts are benchmark-agnostic, not ad hoc, and emphasize they used the same verdict prompt across LM-based methods to reduce prompt-engineering confounds.

Still outstanding: (1) Verifier’s dilemma / recursive reliance on LMs: modularity helps conceptually, but the reported system still depends heavily on LM substeps, and the paper’s own error analysis acknowledges verifier mistakes—so reliability remains a limitation rather than solved. (2) Component-level diagnostics: they justify why “evidence-selection accuracy” is hard to label at scale, but reviewers’ desire for stronger diagnostics remains; the ablations help, yet there’s still no direct measurement of decomposition/evidence quality or sensitivity analysis when those upstream steps fail. (3) Scalability in latency-sensitive deployments: even with cheaper models and smaller hyperparameters, VeriTrail remains substantially slower/more expensive than lightweight baselines; the rebuttal explains the tradeoff but doesn’t eliminate it—so the method’s practical positioning should remain clearly scoped to settings where auditability/traceability is worth the overhead.

**Reviewer Scores:**

Reviewer 1wLF (score 8): Likely unchanged at 8. Their main asks were intermediate evidence-selection evaluation and API/token-cost comparisons; the authors clarified cost analysis exists and justified why evidence-selection “accuracy” is hard to label, which should largely satisfy them.

Reviewer xoWU (score 6): Likely unchanged at 6. The rebuttal helps by emphasizing verifier modularity and transparency, and by equalizing the verdict prompt across methods; however, the core “verifier’s dilemma” concern is structural and probably remains, limiting how far they’d raise.

Reviewer 8wrP (score 6): Likely 6. They explicitly said they could raise their score if runtime/cost and stage-assignment sensitivity were addressed; the authors provided runtime tables, cost tradeoffs, and explained stage inference and insensitivity, which should move them upward.

Reviewer xtw1 (score 2): Likely up to 4. Several of their strongest criticisms appear to stem from misunderstandings the authors directly corrected (scope beyond summarization/QA included; a DAG-using baseline/ablation exists; prompts intended to be benchmark-agnostic). They may still dislike prompt design complexity and overhead, but those are typically “limitations” rather than reasons to stay at reject once misunderstandings are cleared.

---

### Decision · Program_Chairs · 2026-01-26

Accept (Poster)